# ENSUR: Equitable and Statistically Unbiased Recommendation

**Nitin Bisht**[1]  **Xiuwen Gong**[1]  **Guandong Xu**[2]

## Abstract

Although Recommender Systems (RS) have been well-developed for various fields of applications, they often suffer from a crisis of platform credibility with respect to RS confidence and fairness, which may drive users away, threatening the platform's long-term success. In recent years, some works have tried to solve these issues; however, they lack strong statistical guarantees. Therefore, there is an urgent need to solve both issues with a unifying framework with robust statistical guarantees. In this paper, we propose a novel and reliable framework called Equitable and Statistically Unbiased Recommendation (ENSUR)) to dynamically generate prediction sets for users across various groups, which are guaranteed 1) to include ground-truth items with user-predefined high confidence/probability (e.g., 90%); 2) to ensure user fairness across different groups; 3) to have minimum efficient average prediction set sizes. We further design an efficient algorithm named Guaranteed User Fairness Algorithm (GUFA) to optimize the proposed method and derive upper bounds of risk and fairness metrics to speed up optimization process. Moreover, we provide rigorous theoretical analysis concerning risk and fairness control and minimum set size. Extensive experiments validate the effectiveness of the proposed framework, which aligns with our theoretical analysis.

## 1. Introduction

Recommender Systems (RS) (Aggarwal, 2016; Fan et al., 2022; Sharma et al., 2024) are a type of information filtering system designed to provide suggestions to users based on their preferences. While much effort goes into improving accuracy of these recommendation models, less attention has been paid to model confidence, affecting users' trust in the platform's credibility. In recent years, few recommendation approaches (Naghiaei et al., 2022; KWEON et al., 2024) are developed for model confidence. However, these methods are heuristic modeling without statistical guarantee. Meanwhile, fairness is another critical issue that may harm user experience and undermine platform reliability. Some fairness-based recommendation models have been developed in recent years (Li et al., 2023; Han et al., 2024). While these papers alleviate fairness issues in recommendation systems, they are typically empirically validated without statistical guarantees for both performance and fairness.

As a result, we are motivated to develop a complete and statistically guaranteed recommendation framework that considers both model confidence and fairness issues as a whole in this paper. Our overall goal is to construct set predictors that can generate minimum prediction set for each user while guaranteeing model confidence and ensuring user fairness among different groups. Thus, objectives of our framework are threefold: (1) to construct prediction sets that cover true item with high user pre-defined probability, say 90% (i.e., confidence level); (2) to guarantee user fairness across different groups; and (3) to guarantee minimum average set size while ensuring (1) and (2).

Inspired by Risk-Controlling Prediction Sets (RCPS) (Bates et al., 2021b) - a powerful statistical tool, we propose a reliable and fair framework called Equitable and Statistically Unbiased Recommendation (ENSUR)) to achieve the above-mentioned objectives. However, RCPS in its natural form, is designed only to ensure coverage guarantees. As a result, it does not address our key objectives, specifically: 1) how to guarantee fairness among different user groups definitions in a statistical way? 2) how to improve the efficiency of constructing prediction sets when the search range is so large? 3) how to produce recommendation sets with minimum size? 4) how to theoretically guarantee the constructed prediction sets meet the risk control and fairness definition as well as minimum set size? To address these gaps, we first define an estimator called fairness metric, which is required to meet the Fairness-Controlling Prediction Sets (FCPS) defined in a similar way as that of risk control. We then build our objective function by minimizing average prediction set while making it meet both RCPS and FCPS constraints for all users across different groups. Subsequently, we derive upper bounds for both the risk and fairness to accelerate

---

[1]University of Technology, Sydney [2]The Education University of Hong Kong. Correspondence to: Xiuwen Gong <Xiuwen.Gong@uts.edu.au>, Guandong Xu <gdxu@eduhk.hk>.

*Proceedings of the 42nd International Conference on Machine Learning*, Vancouver, Canada. PMLR 267, 2025. Copyright 2025 by the author(s).

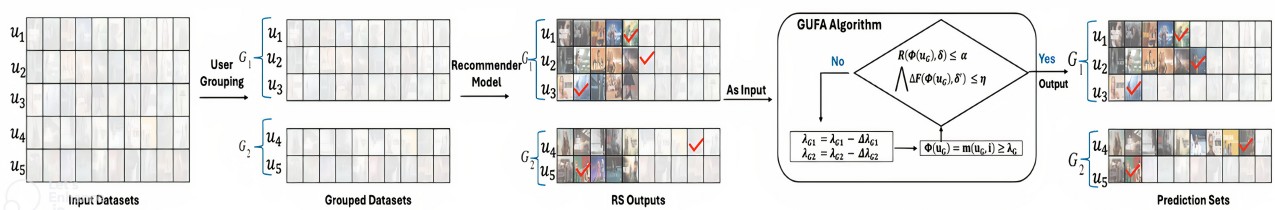

*Figure 1.* The proposed ENSUR Framework. Red check marks indicate the true relevant items.

optimization process for prediction set construction. Lastly, we provide theoretical analysis to prove effectiveness of set predictors with respect to RCPS and FCPS, and minimum set size. The proposed framework is depicted in Figure 1.

Our contributions are summarized as follows:

- Firstly, we formulate the recommendation problem from statistically guaranteed perspectives in terms of risk control guarantee and fairness control guarantee, and propose a reliable and fair recommendation framework, i.e., Equitable and Statistically Unbiased Recommendation (ENSUR)), which is able to construct minimum prediction set while ensuring the risk control and fairness guarantee for all users in different groups.
- Secondly, we design an efficient optimization algorithm, i.e., Greedy User Fairness Algorithm (GUFA) to optimize the objective function of ENSUR. To accelerate the optimization process, we derive the upper bounds for both the defined expected risk and fairness metric via concentration inequalities in Theorem 4.1 and Theorem 4.2 and then make them approach their respective thresholds in a greedy way.
- Next, we establish rigorous theoretical guarantees for the proposed framework ENSUR. We prove that the constructed prediction set can achieve risk control and fairness guarantees in Theorem 5.1 while achieving minimal set sizes in Theorem 5.2, which theoretically verifies the effectiveness of ENSUR.
- Finally, we conduct comprehensive experiments on top of five commonly used recommendation models and various datasets across multiple domains and fairness definitions, demonstrating the empirical efficiency and effectiveness of the proposed ENSUR, which also aligns with our theoretical analysis.

## 2. Related Works

### 2.1. Recommendation

Recommender systems (RS) (Ko et al., 2022; Lu et al., 2015) help users make decisions via personalized content in different fields of application, such as e-commerce (Schafer

---

The code and implementation details are available at https://github.com/kalpiree/ENSUR

et al., 1999), media streaming (Chang et al., 2017), social networks (He et al., 2024) etc. Credibility and fairness are two crucial factors in ensuring the satisfaction of customers and the long-term success of these systems. Traditional recommendation models primarily focused on accuracy (Adomavicius & Tuzhilin, 2005; Ricci et al., 2010), however, aligning with broader trends in machine learning (Huang et al., 2021; Liu et al., 2019; Zou & Liu, 2023), there is a growing appreciation that model confidence, the reliability of a recommendation, is equally important. However, most of these methods are heuristic modeling without statistical guarantees (Naghiaei et al., 2022). Meanwhile, some fairness-based recommendation models have been developed in recent years, which usually focus on a particular fairness issue in specific fields of application. Fairness in RS can be viewed from diverse perspectives (Li et al., 2023). One such perspective is Individual fairness and Group fairness. Individual fairness requires that similar individuals receive comparable treatment. However, defining this similarity is challenging due to disagreements over task-specific similarity metrics (Dwork et al., 2011). Group fairness, on the other hand, ensures that protected groups receive treatment comparable to that of advantaged groups or the general population (Pedreschi et al., 2009), thus ensuring equitable treatment across predefined groups. It can be further classified from the user side or item/platform side. Focusing on User-Side group Fairness, it can be defined based on sensitive features like age, gender, race, etc. (Yao & Huang, 2017) utilized gender to distinguish between advantaged and disadvantaged user groups and measured prediction discrepancies. Another approach utilizes differentiating groups based on user interactions as defined by (Li et al., 2021a) and (Abdollahpouri et al., 2019). To ensure fairness, existing works apply several techniques such as regularization and constrained optimization (Li et al., 2021a; Islam et al., 2021). Some other approaches use Reinforcement Learning by formulating the problem as a Constrained Markov Decision Process (Ge et al., 2021; 2022). To evaluate the fairness, (Yao & Huang, 2017) introduced four group metrics to evaluate collaborative filtering recommender models. (Fu et al., 2020) employed the Group Recommendation Unfairness (GRU) metric to assess disparities across these user groups. Rahmani et al. (2022) depicted this approach balances fairness with utility under certain conditions. Unlike

the more robust statistical frameworks utilized in general machine learning works (Gong et al., 2023b;a; 2021), these fairness methods do not have a notion of statistical guarantees. Addressing that gap is focus of our work.

## 2.2. Risk-Controlling Prediction Sets

We develop uncertainty quantification for the model confidence and fairness based on Risk-Controlling Prediction Sets (RCPS) (Bates et al., 2021b). RCPS is a general framework, not a specific algorithm, for producing predictive sets that satisfy the risk control in Definition 1. Different contexts require different designs of risk or other estimators to achieve best performance. For example, in the context of medical diagnosis, if set $S(X)$ represents plausible diagnoses based on patient features $X$ and $R(S)$ is expected risk of loss from missing true diagnoses, then RCPS ensures this risk to remain below $\alpha$ with confidence $1 - \delta$. This enables doctors to automatically screen for many diseases (e.g., via a blood sample) and refer the patient to relevant specialists. We will apply framework of RCPS to the designed risk and fairness in the context of recommendation.

*Definition* 1 (Risk-controlling prediction sets (RCPS) (Bates et al., 2021b)). Let $\mathcal{S}$ be a random function taking values in space of functions $\mathcal{X} \to \mathcal{Y}'$ (e.g., a functional estimator trained on data). We say that $\mathcal{S}$ is a $(\alpha, \delta) - RCPS$ if, with probability at least $1 - \delta$, we have $\mathcal{R}(\mathcal{S}) \leq \alpha$.

## 3. The Proposed Framework

In this section, we formulate the objective functions that our framework, i.e., Equitable and Statistically Unbiased Recommendation (ENSUR)), aims to achieve. Firstly, we introduce the notations used in the paper. Consider $n$ items, denoted as $\boldsymbol{i} = [i]_{j=1}^{n}$, where each item $i_j$ is an element of the item space $\mathcal{I}$. Similarly, we have $m$ users, represented by $\boldsymbol{u} = [u]_{k=1}^{m}$, where each user $u_k$ belongs to the user space $\mathcal{U}$. For brevity, we use $u$ and $i$ for user and item, respectively. The group information $G$ of each user $u$ is known, and following (Li et al., 2021b), we partition users into two groups, $G_1$ and $G_2$, such that $G_1 \cap G_2 = \emptyset$ and $G_1 \cup G_2 = \mathcal{U}$ to ensure exclusivity. Here, $G_1$ and $G_2$ represent the advantaged and disadvantaged groups, respectively.

The recommendation is conducted via relevance model $m :$ $\mathcal{U} \times \mathcal{I} \to [0, 1]$, which maps a user $u$ and an item $i$ to an estimate score $m(u, i)$, and items with the highest scores are usually the most relevant recommendations. However, there is no theoretical guarantee to ensure the confidence of the model's output, and so the reliability of the recommended items remains uncertain. In the following, we will follow the framework of Risk-Controlling Prediction Sets (RCPS) Bates et al. (2021a) to solve this gap. We define our set predictor to be $\phi : u \to \boldsymbol{i}'$, where $\boldsymbol{i}' \subseteq \mathcal{I}$ is a set-valued output guided by parameter $\lambda$. This lambda takes values in

a closed set $\Lambda \subset \mathbb{R}$ such that $\phi(.)$ is nested i.e.,

$$\lambda_1 < \lambda_2 \implies \phi_{\lambda_2}(u) \subset \phi_{\lambda_1}(u). \tag{1}$$

Considering the recommendation setting with implicit feedback (Hu et al., 2008; Zhu et al., 2024), we define the loss function between the relevant item $i_{\text{true}}$ of user $u$ and the prediction set $\phi_\lambda(u)$ to be 0-1 loss as follows:

$$L(i_{\text{true}}, \phi_\lambda(u)) = \begin{cases} 1 & \text{if } i_{\text{true}} \notin \phi_\lambda \\ 0 & \text{if } i_{\text{true}} \in \phi_\lambda. \end{cases} \tag{2}$$

Using (Bates et al., 2021b), the loss function $L(i_{\text{true}}, \phi_\lambda(u))$ is assumed to also satisfy the following property:

$$\phi_{\lambda_1}(u) \subset \phi_{\lambda_2}(u) \implies L(i_{\text{true}}, \phi_{\lambda_1}(u)) \geq L(i_{\text{true}}, \phi_{\lambda_2}(u)). \tag{3}$$

Based on the above loss function, we define the expected risk of not including a ground-truth item in the prediction set for all users as follows:

$$R(\lambda_G) = E(L(i_{\text{true}}, \phi_{\lambda_G}(u))). \tag{4}$$

Subsequently, we require defined risk to meet risk-controlling prediction sets (RCPS), which ensures the probability of risk lower than user-specified threshold $\alpha$ is no less than user-defined confidence level $1 - \delta$, namely, reliability of recommendation. This can be formulated as follows:

$$\Pr(R(\lambda_G) \leq \alpha)) \geq 1 - \delta. \tag{5}$$

Meanwhile, fairness among users in the advantaged groups and the disadvantaged groups is another challenge that needs to be tackled. Notably, in recommendation settings, "advantaged" or "disadvantaged" can stem from various factors—such as demographics, engagement patterns, or other domain-specific attributes. Thus, we define a fairness metric $\Delta F(\cdot)$ via the difference between the normally used recommendation metric (such as hit rate (HR) and DCG) of the advantaged group $G_1$ and the disadvantaged group $G_2$, to evaluate user fairness as follows:

$$\Delta F(\lambda_{G_1}, \lambda_{G_2}) := \left| \frac{1}{|G_1|} \sum_{u \in G_1} M(\phi_{\lambda_{G_1}}(u)) - \frac{1}{|G_2|} \sum_{u \in G_2} M(\phi_{\lambda_{G_2}}(u)) \right|. \tag{6}$$

Here, $M(\cdot)$ denotes generalized function representing recommendation metric (such as HR or DCG) that measures performance of recommendation set $\phi_{\lambda_G}(u)$ for any user $u$.

For example, when we use hit rate (HR) or DCG as the recommendation metric, we can express them as:

$$\text{HR}(G_i) = \frac{1}{|G_i|} \sum_{u \in G_i} \mathbb{I}(\text{relevant item in } \phi_{\lambda_{G_i}}(u)),$$

$$\mathrm{DCG}(G_i) = \frac{1}{|G_i|} \sum_{u \in G_i} \mathrm{DCG}.(\phi_{\lambda_{G_i}}(u)),$$

Thus, the fairness metrics can be expressed as:

$$\Delta_{\mathrm{HR}} = |\mathrm{HR}(G_1) - \mathrm{HR}(G_2)|,$$
$$\Delta_{\mathrm{DCG}} = |\mathrm{DCG}(G_1) - \mathrm{DCG}(G_2)|.$$

This design makes our proposed framework more flexible by accommodating different types of RS metrics and diverse user-group definitions.

Similarly, we require the defined fairness metric to meet the fairness-controlling prediction sets (FCPS), that is, the probability of the fairness metric lower than a user-specified threshold $\eta$ is no less than user-pre-defined confidence level $1 - \hat{\delta}$, namely, the reliability of fairness. The detailed formulation can be expressed as follows:

$$\Pr(\Delta F(\lambda_{G_1}, \lambda_{G_2}) \leq \eta) \geq 1 - \hat{\delta}. \tag{7}$$

Moreover, we hope constructed prediction sets to be as small as possible while they meet the risk-controlling guarantee as well as the fairness-controlling guarantee. This is because a smaller but more relevant set not only reduces uncertainty in recommendations (Coscrato & Bridge, 2023) but also enhances user satisfaction and eases cognitive load (Chen et al., 2022), ultimately improving the usability and effectiveness of the RS. Therefore, our goal is to find the optimal $(\lambda_{G_1}, \lambda_{G_2})$ that minimizes the average size of the recommendation sets, satisfying the risk (coverage) and fairness guarantees for all users in groups $G_1$ and $G_2$. The objective function can be formulated as follows:

$$\underset{(\lambda_{G_1}, \lambda_{G_2})}{\arg\min} \sum_{G \in \{G_1, G_2\}} \frac{1}{|G|} \sum_{u \in G} |\phi_{\lambda_G}(u)|$$
$$\text{s.t.} \quad \Pr(R(\lambda_G) \leq \alpha)) \geq 1 - \delta \text{ for all } G \in \{G_1, G_2\},$$
$$\Pr(\Delta F(\lambda_{G_1}, \lambda_{G_2}) \leq \eta) \geq 1 - \hat{\delta}. \tag{8}$$

Here, $\alpha$ and $\eta$ are the user pre-specified risk and fairness thresholds, say 10%; $1 - \delta$ and $1 - \hat{\delta}$ are the user pre-defined confidence level for the risk and fairness, say 90%.

## 4. The Optimization Algorithm

To optimize the objective function in Equation (8), we need to ensure the risk and fairness metric in the constraints are below decision-makers' pre-defined value $\alpha$ and $\eta$ respectively, and finally obtain the optimal prediction set with minimum size. It is not efficient to directly apply the greedy algorithm as the range of risk and fairness values that approach the threshold $\alpha$ and $\eta$ by adjusting the $(\lambda_{G_1}, \lambda_{G_2})$ is very large. If we can derive the upper bounds of both risk and fairness metric and take values at their corresponding

upper bounds $R_G^+(\lambda_G, \delta)$ and $\Delta F^+(\lambda_{G_1}, \lambda_{G_2}, \hat{\delta})$ respectively, then it will become more efficient to approach the threshold $\alpha$ and $\eta$ by adjusting the $(\lambda_{G_1}, \lambda_{G_2})$. Following the upper bound strategy to accelerate the optimization procedures in (Bates et al., 2021b), we have the optimized risk constraint as follows:

$$\Pr(R(\lambda_G) \leq R^+(\lambda_G, \delta)) \geq 1 - \delta$$
$$\text{and} \quad R_G^+(\lambda_G, \delta) \leq \alpha, \quad \text{for all } G \in \{G_1, G_2\}. \tag{9}$$

Similarly, the optimized fairness metric constraint can be reformulated as follows:

$$\Pr(\Delta F(\lambda_{G_1}, \lambda_{G_2}) \leq \Delta F^+(\lambda_{G_1}, \lambda_{G_2}, \hat{\delta})) \geq 1 - \hat{\delta}$$
$$\text{and} \quad \Delta F^+(\lambda_{G_1}, \lambda_{G_2}, \hat{\delta}) \leq \eta. \tag{10}$$

Consequently, we can choose $\hat{\lambda}$ as the largest value of $\lambda$ such that the entire confidence region to the left of $\lambda$ falls below the target risk level $\alpha$ and $\eta$, and the set size will achieve the minimum value. The optimized objective function can be formulated as follows:

$$(\hat{\lambda}_{G_1}, \hat{\lambda}_{G_2}) = \sup \Big\{ \lambda_{G_1}, \lambda_{G_2} \in [0, 1] :$$
$$R^+(\lambda_G, \delta) \leq \alpha, \Delta F^+(\lambda_{G_1}, \lambda_{G_2}, \hat{\delta}) \leq \eta \Big\}. \tag{11}$$

To optimize the above objective function and output the optimal solution for $(\hat{\lambda}_{G_1}, \hat{\lambda}_{G_2})$ that dominate the validity of set predictor, we design a novel greedy-strategy-based algorithm called Greedy User Fairness Algorithm (GUFA). The complete procedures of the optimization algorithm are summarized in Algorithm 1.

However, it still remains unknown that what the upper bounds of risk and fairness metric look like. In the following part, we will derive the upper bounds in Theorem 4.1 and Theorem 4.2 respectively.

**Theorem 4.1** ( Upper Bound for Risk). *Assume loss function* $L(i_{true}, \phi_{\lambda_G}(u))$ *follows a Bernoulli distribution, then upper bound for the risk* $R(\lambda_G)$ *can be found as follows:*

$$R^+(\lambda_G, \delta) = \sup \Big\{ \hat{R}(\lambda_G) : BinomCDF(n\hat{R}(\lambda_G), n, \alpha) \leq \delta \Big\} \tag{12}$$

*where* $n$ *is the number of samples;* $G \in \{G_1, G_2\}$; $\hat{R}(\lambda_G)$ *denotes the empirical risk of* $R(\lambda_G)$, *which can be calculated as follows:*

$$\hat{R}(\lambda_G) = \frac{1}{|G|} \sum_{u \in G} L(i_{true}, \phi_{\lambda_G}(u)). \tag{13}$$

*Here,* $|G|$ *denotes the number of users in group* $G$.

*Proof.* Proof can be found in Appendix B.1. $\qquad \square$

**Algorithm 1** Guaranteed User Fairness Algorithm (GUFA)

1: **Initialization:**
2: Initialize control parameters for two groups $\lambda_{G_1}, \lambda_{G_2}$
3: Initialize user pre-specified parameters $\alpha$, $\eta$, $\delta$, $\hat{\delta}, \Delta_1, \Delta_2$
4: **Define** Loss as in Equation (2)
5: **Define** Fairness metric as in Equation (6)
6: **Adjustment Loop:**
7: **for** users in each group $G \in \{G_1, G_2\}$ **do**
8:     Calculate $R_G^+(\lambda_G, \delta)$ such that $\Pr(R(\lambda_G) \leq R_G^+(\lambda_G, \delta)) \geq 1 - \delta$
9:     Compute $\Delta F(\lambda_{G_1}, \lambda_{G_2})$ and calculate $\Delta F^+(\lambda_{G_1}, \lambda_{G_2}, \hat{\delta})$ such that $\Pr(\Delta F(\lambda_{G_1}, \lambda_{G_2}) \leq \Delta F^+(\lambda_{G_1}, \lambda_{G_2}, \hat{\delta})) \geq 1 - \delta$
10:     **if** $R_G^+(\lambda_G, \delta) > \alpha$ OR $\Delta F^+(\lambda_{G_1}, \lambda_{G_2}, \hat{\delta}) > \eta$ **then**
11:         Update $\lambda_{G_1} \leftarrow \lambda_{G_1} - \Delta_1, \lambda_{G_2} \leftarrow \lambda_{G_2} - \Delta_2$
12:     **end if**
13: **end for**
14: $\hat{\lambda}_{G_1}, \hat{\lambda}_{G_2} \leftarrow \lambda_{G_1}, \lambda_{G_2}$    ▷ Get the optimal $\lambda_{G_1}, \lambda_{G_2}$
15: **Construct Prediction Sets:**
16: **for** each user $u$ in group $G$ **do**
17:     $\phi_{\hat{\lambda}_G}(u) \leftarrow \{i \mid m(u, i) \geq \hat{\lambda}_G\}$
18: **end for**
19: **Output:** the optimal solution $\hat{\lambda}_{G_1}, \hat{\lambda}_{G_2}$ and prediction sets $\phi_{\hat{\lambda}_{G_1}}(u)$ and $\phi_{\hat{\lambda}_{G_2}}(u)$ for all users in different groups.

---

**Theorem 4.2** (Upper Bound for Fairness Metric). *The upper bound for fairness metric $\Delta F(\lambda_{G_1}, \lambda_{G_2})$ can be derived by applying Bernstein inequality (Maurer & Pontil, 2009) as follows:*

$$\Delta F^+(\lambda_{G_1}, \lambda_{G_2}, \hat{\delta}) =$$
$$\Delta F(\lambda_{G_1}, \lambda_{G_2}) + \sqrt{\frac{2\sigma_F^2 \log\left(\frac{2}{\hat{\delta}}\right) + \frac{2}{3}\log\left(\frac{2}{\hat{\delta}}\right)}{n_1 + n_2}}. \quad (14)$$

*where $n_1$ and $n_2$ denote the number of samples for group $G_1$ and $G_2$; $\sigma_F^2$ denotes the variance associated with the fairness metric $\Delta_{HR}$ or $\Delta_{DCG}$. The detailed formulation of the variance can be referred to in Appendix A.1.*

*Proof.* Proof can be found in Appendix B.2. □

**Recommendation**   After obtaining optimal $(\hat{\lambda}_{G_1}, \hat{\lambda}_{G_2})$ from Algorithm 1, we can recommend new items for users. For example, when user $u$ comes, we first decide on group $G$ that they belong to and then utilize corresponding $\hat{\lambda}_G$ to calculate their prediction set via step 17.

## 5. Theoretical Analysis

In this section, we provide theoretical analysis on the risk and fairness control guarantee in Theorem 5.1, as well as the minimum set size guarantee in Theorem 5.2.

**Theorem 5.1** (Risk and Fairness Control Guarantee). *For all group $G \in \{G_1, G_2\}$ and $\delta \in (0, 1)$, with probability of at least $1 - \delta$ for risk threshold $\alpha$, and with probability of at least $1 - \hat{\delta}$ for fairness threshold $\eta$, we have:*

$$\Pr(R(\hat{\lambda}_G) \leq \alpha) \geq 1 - \delta$$
$$\wedge \Pr(\Delta F(\hat{\lambda}_{G_1}, \hat{\lambda}_{G_2}) \leq \eta) \geq 1 - \hat{\delta}. \quad (15)$$

*Proof.* Proof can be found in Appendix B.3. □

**Remark.**   In Theorem 5.1, we prove that the optimal $\hat{\lambda}_{G_1}, \hat{\lambda}_{G_2}$ obtained from Algorithm 1 are indeed able to control the expected risk to below the decision makers' defined values of $\alpha$ with confidence $1 - \delta$, and control the fairness metric $\Delta F$ to below the decision makers' defined values of $\eta$ with confidence $1 - \hat{\delta}$. This theoretically validate the recommendation reliability and fairness of the proposed ENSUR framework.

**Theorem 5.2** (Minimum Set Size Guarantee). *Let $(\phi_{\lambda_{G_1}^*}, \phi_{\lambda_{G_2}^*})$ be any set predictor and let $(\phi_{\hat{\lambda}_{G_1}}, \phi_{\hat{\lambda}_{G_2}})$ be the optimal predictor obtained from Algorithm 1 such that $R(\lambda_G^*) \leq R(\hat{\lambda}_G)$ and $\Delta F(\lambda_{G_1}^*, \lambda_{G_2}^*) \leq \Delta F(\hat{\lambda}_{G_1}, \hat{\lambda}_{G_2})$. Then for each $G \in \{G_1, G_2\}$, we have:*

$$E\left[|\phi_{\hat{\lambda}_G}(u)|\right] \leq E\left[|\phi_{\lambda_G^*}(u)|\right]. \quad (16)$$

*where $|\phi_{\hat{\lambda}_G}(u)|$ denotes the predicted set size for any user $u$ in group $G$.*

*Proof.* Proof can be found in Appendix B.4. □

**Remark.**   In Theorem 5.2, we prove that set predictor learned by our algorithm can output the minimal prediction set size for any user $u$ in group $G$, which theoretically validate the effectiveness of the proposed ENSUR framework.

To sum up, set predictors constructed by Algorithm 1 can modify any black-box recommendation models to output prediction sets for new customers that are strictly guaranteed to satisfy the risk control as defined in Equation (5) and the fairness control defined in Equation (7) while ensuring the minimum prediction sets in Equation (8).

## 6. Experiments

In this section, we conduct experiments to validate the effectiveness of the proposed framework (ENSUR). We design

experiments to 1) validate whether the framework can provide desired coverage guarantee in terms of risk, better performance in terms of average set size, and improved fairness in terms of Hit Rate Difference (Hit Rate Diff.) and DCG Difference (DCG Diff.) across various datasets with sensitive attributes; 2) analyze how the parameters ( i.e. $\alpha$, $\delta$, $\eta$ and $\hat{\delta}$) influence the performance; 3) analyze the time-efficiency of ENSUR compared to other fairness baselines.

## 6.1. Datasets and Base Models

We conduct experiments on four datasets with specific sensitive user attributes: (1) AmazonOffice dataset (eCommerce) (McAuley et al., 2015) grouped by item interactions; (2) Last.fm dataset (music streaming) (Cantador et al., 2011) grouped by region (developed and other countries; (3) MovieLens dataset (movie ratings) (Harper & Konstan, 2015) grouped by gender; and (4) Book-Crossing dataset (book ratings) (Ziegler et al., 2005) grouped by age. We implement the proposed framework on five base recommendation models: DeepFM (Guo et al., 2017), GMF (Koren et al., 2009), MLP (Zhang et al., 2019), NeuMF (He et al., 2017), and LightGCN (He et al., 2020). Additionally, we compare our framework ENSUR with four fairness baselines: 1) NFCF (Islam et al., 2021) 2) MFCF (Islam et al., 2021) 3) GMF-UFR (Li et al., 2021a) 4) NCF-UFR (Li et al., 2021a). The implementation details and the details of all the datasets, base models, and fairness baselines can be found in Appendices C and D.

## 6.2. Experimental Results

### 6.2.1. RESULTS W.R.T PERFORMANCE AND FAIRNESS

We compare the performance and fairness of the ENSUR framework with five base recommendation models and four fairness baselines. We set the predefined risk threshold $\alpha = 0.20$, fairness threshold $\eta = 0.20$ via manual validation. The error rates $\delta = 0.1$ and $\hat{\delta} = 0.1$ are representatively set following (Bates et al., 2021b). The coverage guarantee is measured in terms of risk; performance is measured using average set size, and fairness is compared using disparity in these metrics between user groups (Difference in Hit Rate and Difference in DCG). The results for the AmazonOffice dataset (grouped by interactions) are provided in Table 1 whereas results for MovieLens dataset (grouped by gender), Last.fM dataset (grouped by region), and Book-Crossing dataset (grouped by age) are provided in Tables 3 to 5 respectively in Appendix E.

The results, presented in Table 1 lead us to the following key observations:

- The ENSUR framework ensures that all base models generate prediction sets that satisfy both risk control

*Table 1.* Performances and fairness comparisons with base models and fairness baselines on **AmazonOffice Dataset** grouped by the **Interactions** in terms of risk, average set size, and Hit Rate Diff/DCG Diff, respectively. Bold indicates best result, underline indicates the second best and † marks threshold exceeded cases.

| Method | Group | Risk ↓ | Average Set Size ↓ | Hit Rate | DCG | Hit Rate Diff ↓ | DCG Diff ↓ |
|---|---|---|---|---|---|---|---|
| DeepFM | 1 | 0.121 | 34 | 0.879 | 0.418 | 0.155 | 0.17 |
|  | 2 | 0.277 † |  | 0.723 | 0.248 |  |  |
| DeepFM + ENSUR | 1 | 0.192 |  | 0.808 | 0.401 | 0.081 | **0.103** |
|  | 2 | 0.111 |  | 0.889 | 0.298 |  |  |
| GMF | 1 | 0.149 | 30 | 0.851 | 0.439 | 0.212 † | 0.225 † |
|  | 2 | 0.361 † |  | 0.639 | 0.214 |  |  |
| GMF + ENSUR | 1 | 0.197 |  | 0.803 | 0.428 | 0.08 | 0.168 |
|  | 2 | 0.117 |  | 0.883 | 0.26 |  |  |
| LightGCN | 1 | 0.077 | 33 | 0.923 | 0.477 | 0.126 | 0.238 † |
|  | 2 | 0.203 † |  | 0.797 | 0.239 |  |  |
| LightGCN + ENSUR | 1 | 0.087 |  | 0.913 | 0.474 | 0.087 | 0.198 |
|  | 2 | 0 |  | 1 | 0.276 |  |  |
| MLP | 1 | 0.162 | **26** | 0.838 | 0.409 | 0.219 † | 0.19 |
|  | 2 | 0.38 † |  | 0.62 | 0.219 |  |  |
| MLP + ENSUR | 1 | 0.197 |  | 0.803 | 0.397 | **0.013** | 0.14 |
|  | 2 | 0.184 |  | 0.816 | 0.257 |  |  |
| NeuMF | 1 | 0.155 | 28 | 0.845 | 0.414 | 0.225 † | 0.185 |
|  | 2 | 0.379 † |  | 0.621 | 0.229 |  |  |
| NeuMF + ENSUR | 1 | 0.182 |  | 0.818 | 0.406 | 0.017 | 0.143 |
|  | 2 | 0.199 |  | 0.801 | 0.263 |  |  |
| Other Fairness Baselines |  |  |  |  |  |  |  |
| NFCF | 1 | 0.196 | 28 | 0.804 | 0.391 | 0.115 | 0.134 |
|  | 2 | 0.261 † |  | 0.689 | 0.257 |  |  |
| MFCF | 1 | 0.175 | 30 | 0.825 | 0.402 | 0.128 | 0.154 |
|  | 2 | 0.303 † |  | 0.697 | 0.248 |  |  |
| NeuMF-UFR | 1 | 0.193 | 28 | 0.807 | 0.396 | 0.153 | 0.127 |
|  | 2 | 0.346 † |  | 0.654 | 0.269 |  |  |
| GMF-UFR | 1 | 0.205 † | 30 | 0.795 | 0.395 | 0.133 | 0.157 |
|  | 2 | 0.368 † |  | 0.662 | 0.238 |  |  |

and fairness guarantees across all datasets.

- The ENSUR-enhanced models always meet risk below defined thresholds. For base models, the minimum risk threshold criteria is frequently not met. For example, in the AmazonOffice Dataset, we notice, as depicted by †, that risk thresholds are not met for at least one group, i.e., the disadvantaged group across all the base models. In fairness baselines, we observe criteria are not met for both the groups in most cases across all datasets, which may be because of their emphasis on trading off performance for accuracy.

- We also observe that the ENSUR-enhanced models can get the best results in average set size on all the datasets, but the best model varies among different datasets. For example, MLP + ENSUR achieves the best recommendations in terms of average set size on the AmazonOffice dataset. Similar trends are observed for MovieLens, Last.fM and Book-Crossing datasets as depicted in Tables 3 to 5 in Appendix E.

- All ENSUR-enhanced models meet the fairness threshold for both the Hit Rate Diff and DCG Diff across all datasets. However, the best-performing models vary by dataset. For example, MLP + ENSUR achieves the best fairness on the AmazonOffice dataset under the Hit Rate Diff while DeepFM + ENSUR outperforms all the other models in terms of DCG Diff. Tables 3 to 5 in Appendix E depict similar trends for remaining datasets. In addition, the base models do not always achieve the fairness metrics and exceed the fairness threshold marked by †. Meanwhile, the fairness base-

line models do achieve fairness metrics after sacrificing their accuracy, but they are still inferior to the ENSUR-enhanced models.

- Overall, the ENSUR framework effectively ensures both recommendation performance and fairness while guaranteeing risk control, providing valuable insights for real-world applications. We further discuss the generalizability of grouping strategies and practical applicability in Appendix F and Appendix G respectively.

### 6.2.2. PARAMETER ANALYSIS

We further analyze influence of pre-defined risk-related parameters $\alpha$ and $\delta$ and fairness-related parameters $\eta$ and $\hat{\delta}$ on the prediction sets generated by ENSUR framework.

**Effect of Risk Control Parameters $\alpha$ and $\delta$ on Prediction Set Sizes :** We first evaluate the impact of error rate $\alpha$ varying from 0.10 to 0.50 (in increments of 0.05) on average prediction set sizes under fixed risk confidence thresholds $\delta = 0.05, 0.10, 0.15$ using AmazonOffice dataset, grouped by interactions in Figure 2. It can be easily observed that as $\alpha$ increases, the average set size across all models decreases. The decreasing trend demonstrates the framework's ability to generate valid prediction sets that adapt to the error rate $\alpha$. Similar trends can be observed on remaining datasets, see Figures 6 to 8 in Appendix E.2.

We further evaluate effect of varying risk confidence $\delta$ from 0.10 to 0.50 (in increments of 0.05) on average prediction set sizes under fixed risk thresholds ($\alpha = 0.15, 0.20, 0.25$) using Book-Crossing dataset, grouped by age (see Figure 3). In general, all the models show a decreasing trend which validates effectiveness of the proposed framework. Interestingly, prediction set sizes do not seem to fluctuate much for smaller values of $\delta$, while a decreasing trend occurs with increasing $\delta$. This is because relaxing confidence of risk constraints makes our predictions less conservative, thereby reducing the number of items included in prediction set. Similar phenomenon can be obtained on the other datasets, see Figures 9 to 11 in Appendix E.2.

**Effect of Fairness Control Parameters $\eta$ and $\hat{\delta}$ on Prediction Set Sizes :** We analyze how varying $\eta$, measured by the Hit Rate Diff. and DCG Diff. from 0.10 to 0.50 (in increments of 0.05) on the average prediction set sizes under fixed fairness confidence ($\hat{\delta} = 0.15, 0.20, 0.25$) affects average prediction set sizes, measured on the MovieLens dataset grouped by gender (Figure 4). With increasing $\eta$, the prediction set size decreases, validating model's capacity to have smaller prediction sets for less strict $\eta$ condition. The prediction set sizes usually stabilize after an initial decrease as $\eta$ rises, suggesting that the framework's fairness sensitivity to $\eta$ diminishes beyond a certain point. This offers guidance on selecting appropriate fairness thresholds while maintaining usability. Similar results can be observed on

the other datasets, see Figures 12 to 14 in Appendix E.2.

Finally, we examine trends on average prediction set sizes by varying fairness confidence $\hat{\delta}$ from 0.10 to 0.50 (in increments of 0.05) under fixed fairness thresholds ($\eta = 0.15, 0.20, 0.25$) measured on Last.fm dataset grouped by region (Figure 5). We notice that as value of $\hat{\delta}$ increases, for a given fairness threshold, model becomes less conservative, and hence prediction set size decreases. This phenomenon further validates effectiveness of our framework in balancing between producing tight average prediction set size and ensuring fairness. Similarly, results for other datasets can be found in Figures 15 to 17 in Appendix E.2.

Overall, this parameter analysis guides real-world applications in balancing performance and fairness with confidence guarantees.

### 6.2.3. TIME EFFICIENCY COMPARISON

We analyze the computational cost (training time) of the ENSUR framework in comparison with other fairness baselines. Specifically, for in-processing fairness baselines such as NFCF and MFCF, we consider the fine-tuning step to calculate the training time. For post-processinng fairness baselines such as NeuMF-UFR and GMF-UFR, we take the re-reranking step as the training time. For proposed ENSUR framework, we take the calibration step as the training time. We measure the time of ENSUR, averaged on top of all the base models. Our experiments are conducted via 10-fold cross validation to ensure statistical reliability. The results are presented in Table 2.

From the results, we can observe that our proposed framework ENSUR is significantly more time-efficient than other fairness baselines, which indicates scalability of our method. This is because in-processing methods like NFCF and MFCF involves model refitting which substantially increases the computational cost. By contrast, ENSUR operates independently of the training phase, eliminating this overhead. Additionally, ENSUR is substantially faster than NeuMF-UFR and GMF-UFR, other post-processing methods, because these models involve solving a constrained and complex optimization problem, whereas ENSUR employs a simple yet effective greedy-based algorithm.

*Table 2.* Training time (minutes) comparison of our framework ENSUR with four fairness baselines.

| Dataset | MFCF | NFCF | GMF-UFR | NeuMF-UFR | **ENSUR(Ours)** |
|---|---|---|---|---|---|
| AmazonOffice | 45 | 50 | 25 | 22 | **8** |
| MovieLens | 75 | 90 | 49 | 45 | **12** |
| Last.fm | 50 | 58 | 35 | 30 | **8** |
| Book-Crossing | 110 | 135 | 68 | 65 | **15** |

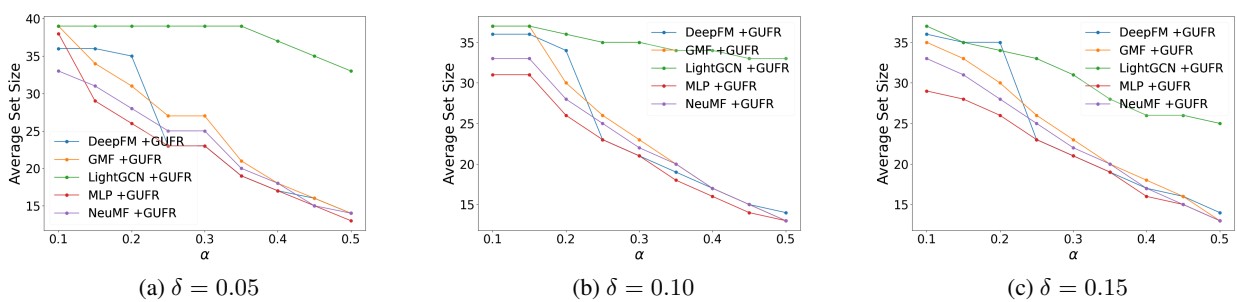

*Figure 2.* Analysis of base models after applying the ENSUR framework in terms of average set size with varying $\alpha = \{0.10, 0.15, 0.20, 0.25, 0.30, 0.35, 0.40, 0.45, 0.50\}$ on **AmazonOffice** dataset grouped by **Interactions** under different $\delta$.

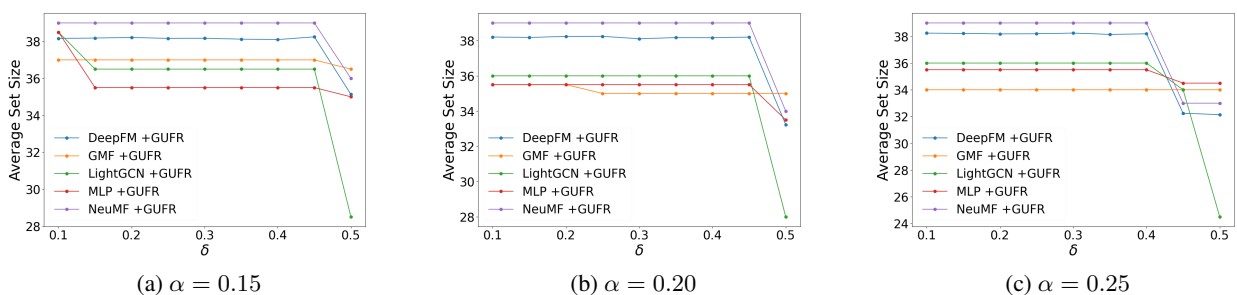

*Figure 3.* Analysis of base models after applying the ENSUR framework in terms of average set size with varying $\delta = \{0.10, 0.15, 0.20, 0.25, 0.30, 0.35, 0.40, 0.45, 0.50\}$ on **Book-Crossing** dataset grouped by **Age** under different $\alpha$.

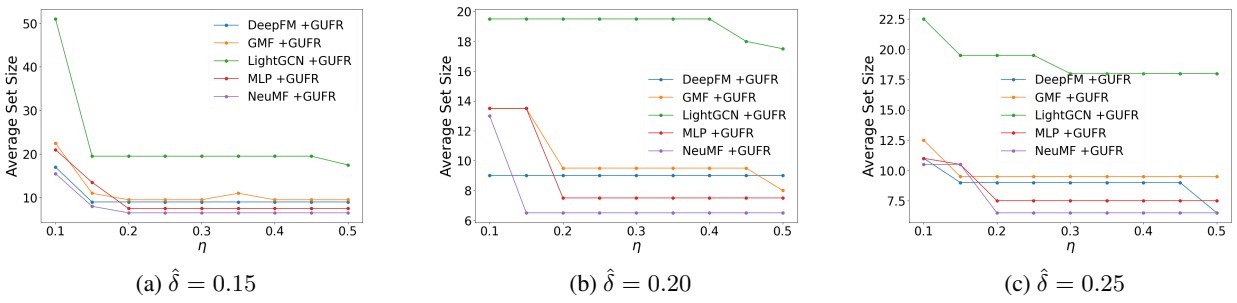

*Figure 4.* Analysis of base models after applying the ENSUR framework in terms of average set size with varying $\eta = \{0.10, 0.15, 0.20, 0.25, 0.30, 0.35, 0.40, 0.45, 0.50\}$ on **MovieLens** dataset grouped by **Gender** under different $\hat{\delta}$.

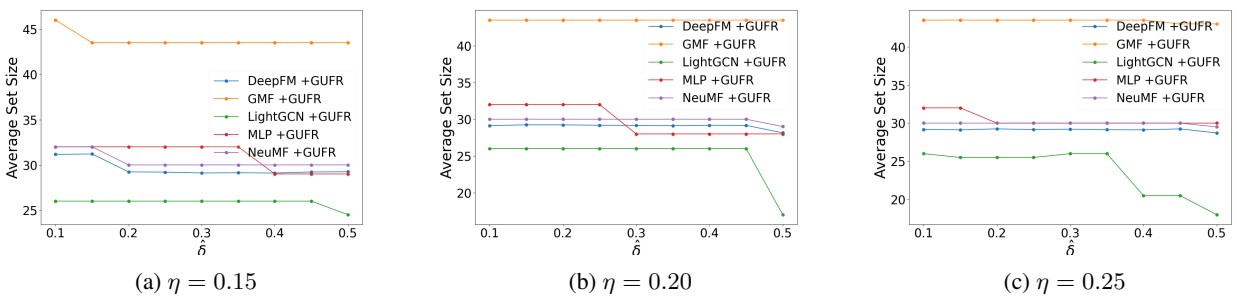

*Figure 5.* Analysis of base models after applying the ENSUR framework in terms of average set size with varying $\hat{\delta} = \{0.10, 0.15, 0.20, 0.25, 0.30, 0.35, 0.40, 0.45, 0.50\}$ on **Last.fm** dataset grouped by **Region** under different $\eta$.

# 7. Conclusion

This paper investigates two principle issues that affect the credibility of RS with respect to confidence and fairness. We integrate the two factors into a unified framework called Equitable and Statistically Unbiased Recommendation (EN-SUR)), which dynamically outputs prediction sets that are guaranteed to have the risk and fairness below a threshold with pre-specified high confidence, such as 90%, while retaining the minimum average size. We conduct theoretical analysis and empirical studies, which are consistent in validating the effectiveness. It is noteworthy that the efficiency of optimizing the ENSUR also depends on the tightness of the derived upper bounds for our risk and fairness, thus, we leave the question whether there exists tighter upper bounds for the future work. Moreover, the proposed framework can work on top of any recommendation model by taking them as black-box, which offers a robust foundation for advancing fairness and reliability in RS, paving the way for future research and development in this field.

## Acknowledgments

This work is partially supported by the Australian Research Council (ARC) Under Grants DP220103717 and LE220100078, and the National Natural Science Foundation of China under Grants No.62072257.

## Impact Statement

Our framework dynamically tailors prediction set sizes in recommender systems, ensuring fairness and performance guarantees while reducing cognitive overload and resource inefficiencies. By addressing disparities in user experiences and promoting equitable recommendations, it advances inclusivity and transparency in user-centric platforms, with applications across e-commerce, streaming, and beyond.

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

# Appendix

## A. Assumptions

**Assumption A.1.** In theorem 4.2, we assume that the groups $G_1$ and $G_2$ are mutually independent and that hit rates or DCG scores are independently distributed within each group. Under these assumptions, the variances for the fairness metrics are calculated as follows:

$$\sigma_{\text{hit}}^2 = \frac{\hat{p}_1(1 - \hat{p}_1)}{n_1} + \frac{\hat{p}_2(1 - \hat{p}_2)}{n_2},$$

$$\sigma_{\text{DCG}}^2 = \frac{s_1^2}{n_1} + \frac{s_2^2}{n_2},$$

where $\hat{p}_1$ and $\hat{p}_2$ are the observed hit rates, and $s_1^2$ and $s_2^2$ are the sample variances of DCG scores for groups $G_1$ and $G_2$, respectively.

This assumption ensures that the application of Bernstein's inequality is valid, allowing us to derive the Upper Confidence Bound (UCB) for fairness metrics as shown in 14.

**Assumption A.2.** Throughout the Theorem 5.1, we make a mild assumption on $\lambda^{\text{min}}$, i.e., the minimum value the parameter $\lambda$ can take, as follows:

$$\Pr(R_G(\lambda_G^{\text{min}}) \leq \alpha) \geq 1 - \delta \wedge \Pr(\Delta F(\lambda_{G_1}^{\text{min}}, \lambda_{G_2}^{\text{min}}) \leq \eta) \geq 1 - \delta$$

where $\lambda_G^{\text{min}}$ is the group-specific minimum value of the parameter for risk control, and $\lambda_{G_1}^{\text{min}}$ and $\lambda_{G_2}^{\text{min}}$ are the minimum values for fairness control across the groups.

This assumption depicts the belief that we can control any user-defined risk $\alpha$ and fairness $\epsilon$ by taking valid $\lambda$ values in a closed set $\Lambda \subseteq \mathbb{R}^2 \cup \{\pm\infty\}$.

## B. Proofs

### B.1. Proof of Theorem 4.1

*Proof.* We focus on finding some $\hat{R}_G^+$ such that out of $n$ samples, $\hat{R}_G$ yields atmost $k = n\hat{R}_G$ successes (where success is defined as observing a risk) with a significance level of atleast $1 - \delta$. The CDF of the binomial distribution is given by:

$$P(\text{Binom}(n, p) \leq k) = \sum_{i=0}^{k} \binom{n}{i} p^i (1-p)^{n-i}.$$

Let us assume we know $\hat{R}_G^+$ and we seek $\hat{R}_G$ such that:

$$P(\text{Binom}(n, \hat{R}_G^+) \leq n\hat{R}_G) \geq 1 - \delta.$$

Replacing $\hat{R}_G^+$ with the user-defined risk value $\alpha$, the equation becomes:

$$P(\text{Binom}(n, \alpha) \leq n\hat{R}_G) \geq 1 - \delta$$

or

$$P(\text{Binom}(n, \alpha) \leq n\hat{R_G}) \leq \delta$$

which can be reformulated as:

$$\text{BinomCDF}(n\hat{R_G}, n, \alpha) \leq \delta.$$

To solve for $\hat{R_G}$, we find the root of this equation which is also the UCB at $\alpha$: i.e.,

$$\text{BinomCDF}(n\hat{R_G}, n, \alpha) - \delta = 0$$

Formally,

$$\hat{R}_G^+ = \sup\left\{\hat{R_G} : \text{BinomCDF}(n\hat{R_G}, n, \alpha) \leq \delta\right\}$$

Hence Proved. □

## B.2. Proof of Theorem 4.2

*Proof.* Bernstein's inequality for a sum of independent random variables $X_i$ with mean $\mu$, variance $\sigma^2$, and bounded by $U$ states:

$$P\left(\left|\frac{1}{n}\sum_{i=1}^{n}(X_i - \mu)\right| \geq t\right) \leq 2\exp\left(-\frac{\frac{1}{2}nt^2}{\sigma^2 + \frac{1}{3}Ut}\right), \tag{17}$$

where $n$ is the number of observations, $X_i$ is the $i$-th random variable, $t$ is the deviation threshold, $\sigma^2$ is the variance of $X_i$, and $U$ is the upper bound on the range of $X_i$.

Analogously, we consider, with some decision-maker confidence value $\hat{\delta}$, that the empirical fairness metric differs from the true fairness metric by the threshold $t$. This can be mathematically represented as:

$$\hat{\delta} = 2\exp\left(-\frac{\frac{1}{2}nt^2}{\sigma_F^2 + \frac{1}{3}Ut}\right),$$

which rearranges to:

$$\log\left(\frac{2}{\hat{\delta}}\right) = \frac{\frac{1}{2}nt^2}{\sigma_F^2 + \frac{1}{3}Ut},$$

solving for $t$ gives:

$$t = \sqrt{\frac{2\sigma_F^2 \log\left(\frac{2}{\hat{\delta}}\right) + \frac{2}{3}U\log\left(\frac{2}{\hat{\delta}}\right)}{n}}.$$

Assuming $U = 1$ conservatively and $n = n_1 + n_2$, we obtain the UCB as:

$$\Delta F^+(\lambda_{G_1}, \lambda_{G_2}, \hat{\delta}) = \Delta F(\lambda_{G_1}, \lambda_{G_2})$$
$$+ \sqrt{\frac{2\sigma_F^2 \log(\frac{2}{\hat{\delta}}) + \frac{2}{3}\log(\frac{2}{\hat{\delta}})}{n_1 + n_2}}.$$

Hence Proved. □

## B.3. Proof of Theorem 5.1

*Proof.* Let $\lambda_G^*$ be the highest parameter value for each group $G \in \{G_1, G_2\}$ such that the expected risk of not including truly relevant items and the fairness metric is less than $\alpha$ and $\eta$ respectively, i.e.,

$$\lambda_G^* = \max\{\lambda_G \in [\lambda_{\min,G}, \lambda_{\max,G}] :$$
$$R_G(\lambda_G) \leq \alpha \wedge \Delta F_G(\lambda_{G_1}, \lambda_{G_2}) \leq \eta\} \tag{18}$$

Assume for a parameter value $\hat{\lambda}_G$, we have $R_G(\hat{\lambda}_G) > \alpha$ or $\Delta F(\hat{\lambda}_{G_1}, \hat{\lambda}_{G_2}) > \alpha$.

Then by the definition of $\lambda_G^*$, we have,

$$R_G(\lambda_G^*) \leq \alpha \wedge \Delta F(\lambda_{G_1}^*, \lambda_{G_2}^*) \leq \eta$$

which implies,

$$R_G(\lambda_G^*) \leq \alpha < R_G(\hat{\lambda}_G) \vee$$
$$\Delta F(\lambda_{G_1}^*, \lambda_{G_2}^*) \leq \eta < \Delta F(\hat{\lambda}_{G_1}, \hat{\lambda}_{G_2})$$

Using Equation (3), we have:

$$\hat{\lambda}_G > \lambda_G^*$$

Since $\hat{\lambda}_G$ and $\lambda_G^*$ are within the range of real numbers, consider some $\xi > 0$ such that

$$(\lambda_G^* + \xi) \geq \hat{\lambda}_G,$$

Utilizing the definition of $\lambda_G^*$ and $\hat{\lambda}_G$ in Equation 18, we get,

$$R_G^+(\lambda_G^* + \xi, \delta) \leq \alpha < R_G(\lambda_G^* + \xi)$$
$$\vee \Delta F^+(\lambda_{G_1}^*, \lambda_{G_2}^* + \xi, \delta) \leq \eta < \Delta F(\lambda_{G_1}^*, \lambda_{G_2}^* + \xi) \tag{19}$$

According to the principles of Upper Confidence Bound (UCB), i.e., eq. 11, the events $R_G^+(\lambda_G^* + \xi, \delta) \leq \alpha$ or $\Delta F^+(\lambda_{G_1}^*, \lambda_{G_2}^* + \xi, \hat{\delta}) \leq \eta$ can only occur with probabilities not exceeding $\delta$ and $\hat{\delta}$ respectively. Specifically, the UCB ensures that the probability of observing $R_G(\hat{\lambda}_G) > \alpha$ is bounded by $\delta$, or the probability of $\Delta F(\hat{\lambda}_{G_1}, \hat{\lambda}_{G_2}) > \eta$ is bounded by $\hat{\delta}$.

Therefore, with complementary probability condition, under Assumption 1 and the defined ranges of $\delta$ and $\hat{\delta}$, we can conclude with confidence that:

$$\Pr(R_G(\hat{\lambda}_G) \leq \alpha) \geq 1 - \delta \quad \wedge$$
$$\Pr(\Delta F(\hat{\lambda}_{G_1}, \hat{\lambda}_{G_2}) \leq \eta) \geq 1 - \hat{\delta}. \tag{20}$$

This validates the assertions of Theorem 3, thereby formally proving the theorem.

□

## B.4. Proof of Theorem 5.2

*Proof.* Since $R_G(\phi_{\lambda_{*,G}}) \leq R_G(\phi_{\hat{\lambda}_G})$ and $\Delta F(\phi_{\lambda_{G_1}, \lambda^*_{G_2}}) \leq \Delta F(\phi_{\lambda_{G_1}, \hat{\lambda}_{G_2}})$, this relationship is expressed through the sum of relevance scores $m(u, i)$ over the items in the respective prediction sets for users:

$$\sum_{u \in G} \sum_{i \in \phi_{\lambda^*_G}(u)} m(u, i) \geq \sum_{u \in G} \sum_{i \in \phi_{\hat{\lambda}_G}(u)} m(u, i),$$

indicating that the accumulated scores of included items in $\phi_{\lambda^*_G}$ are greater.

This is equivalent to:

$$\sum_{u \in G} \sum_{i \in \phi_{\lambda^*_G}(u) \setminus \phi_{\hat{\lambda}_G}(u)} m(u, i) \geq \sum_{u \in G} \sum_{i \in \phi_{\hat{\lambda}_G}(u) \setminus \phi_{\lambda^*_G}(u)} m(u, i).$$

For some items $i \in \phi_{\lambda^*_G}(u) \setminus \phi_{\hat{\lambda}_G}(u)$, $m(u, i) < \hat{\lambda}_G$, and for all items $i \in \phi_{\hat{\lambda}_G}(u) \setminus \phi_{\lambda^*_G}(u)$, $m(u, i) \geq \hat{\lambda}_G$, based on Algorithm 1.

This condition is satisfied if:

$$|\phi_{\lambda^*_G}(u)| \geq |\phi_{\hat{\lambda}_G}(u)|.$$

Thus, the expected size of the set using $\phi_{\hat{\lambda}_G}$ is optimized to be minimal, i.e.,

$$E\left[|\phi_{\hat{\lambda}_G}(u)|\right] \leq E\left[|\phi_{\lambda^*_G}(u)|\right], \tag{21}$$

thereby proving the theorem. $\square$

## C. Implementation Details

All base recommender models are trained for 20 epochs with a batch size of 256, a learning rate of 0.001, the Adam optimizer, and Binary Cross Entropy Loss (BCELoss). For the NFCF and MFCF models, we modified the original code to generalize grouping logic for diverse criteria (e.g., interaction count, age, gender, and geography) and adapted the debiasing process to compute bias directions dynamically for various groups. To ensure consistency, we reused the score files generated by our base models for the GMF-UFR and NeuMF-UFR models. In order to enhance the reproducibility of the results, we utilized MIP (Santos & Toffolo, 2020), a free light-weight Python library for modeling and optimization, instead of Gurobi (Gurobi Optimization, LLC, 2024) optimization solver, a commercially licensed software. For fair and sound comparisons with the base models and fairness baselines, instead of using arbitrary top-k predictions, we utilized the average optimal prediction set size returned by the ENSUR framework on top of the given base recommendation model.

## D. Detailed Experimenation Details

### D.1. Datasets and Grouping Methods

In the main paper, we introduced four user grouping strategies to evaluate the fairness and performance of our framework: (1) grouping based on interaction count with items, (2) grouping based on user age, (3) grouping based on user gender, and (4) grouping based on geographic categorization into developed and other countries. These strategies were applied to the AmazonOffice, Book-Crossing, MovieLens, and Last.fm datasets, respectively. Below, we provide further details on the grouping methodology:

- **Grouping by interaction count:** Following Li et al. (2021a), users were initially evenly split into two groups, with 50% assigned to each group. The groups were then dynamically adjusted to ensure that the minimum interaction count in the advantaged group exceeded the maximum count in the disadvantaged group by at least one.
- **Grouping by age:** Users were divided into two age groups: younger users ($\leq 60$ years) and older users ($> 60$ years).
- **Grouping by gender:** Users were grouped into binary categories based on identified gender (male and female).
- **Geographic categorization:** Users were categorized based on their country of origin into developed (e.g., USA, UK, Europe, Japan etc.) and other countries.

Furthermore, we conducted an additional grouping experiment on the Last.fm dataset. We extended the interaction count-based grouping to incorporate interactions with popular items, following Abdollahpouri et al. (2019). The results of this experiment are provided in Appendix F.

### D.2. Sampling and Data Splitting

We followed the following sampling and splitting method:

- **Negative sampling:** Following Ma et al. (2024), we selected 50 non-interacted items per user through negative sampling for training, validation, and testing.
- **Data splitting:** We employed the Leave-One-Out (LOO) strategy (He et al., 2017; Han et al., 2023) to partition the dataset into training, calibration, and testing sets. Specifically, for each user, one interaction was isolated for calibration and testing, while the remaining interactions were used for training.
- **Multiple trials:** To account for variability in sampling and splitting, we repeated the experiments over 20 independent trials. For each trial, random negative samples were drawn for training, validation, and testing. The results were averaged across all the trials.

## D.3. Model Configurations and Fairness Baselines

To evaluate the effectiveness of our framework, we implemented it on top of the five base recommender models specified in the main paper. Here, we provide specific architectural and training details of the models used:

### Base Recommendation Models

- **DeepFM:** Combines 8 latent factors with deep layers of [50, 25, 10] and ReLU activation.
- **GMF:** Utilizes an embedding size of 8 for capturing linear interactions between user and item embeddings.
- **MLP:** Employs layers of [64, 32, 16] with ReLU activation for modeling non-linear interactions.
- **NeuMF:** Integrates GMF and MLP with a GMF embedding size of 8 and MLP layers of [64, 32, 16], using ReLU activation.
- **LightGCN:** Configured with an embedding size of 8 and 3 graph convolution layers.

To validate our framework further, we compared it with four fairness baseline approaches. The baselines are based on the most commonly adopted methods in fairness literature i.e. in-processing and post-processing methods (Li et al., 2023) :

### Fairness Baselines

- **NFCF and MFCF(In-processing) (Islam et al., 2021):** The authors utilize a pre-training and fine-tuning approach to induce user-sided group fairness. Initially, the user embeddings are learned from non-sensitive interactions, followed by a de-biasing step to mitigate the embedding bias. Finally, the models are fine-tuned on sensitive item recommendations with a fairness penalty to reduce systemic bias in predictions.
- **Neumf-UFR AND GMF-UFR (Post-processing) (Li et al., 2021a):** This post-hoc re-ranking approach utilizes an integer programming solver to balance fairness and utility disparity between advantaged and disadvantaged user groups. The method optimizes preference scores while enforcing a fairness constraint, ensuring that recommendation quality differences (e.g., DCG@10, F1@10) between groups remain below a specified threshold.

# E. Additional Experiments

## E.1. Remaining Experiments -Continued

Tables 3 to 5 extend the analysis provided in the main paper. These tables support the key findings: the ENSUR framework consistently achieves both risk control ($\alpha = 0.20$) and fairness ($\eta = 0.20$) thresholds across all datasets, outperforming base models and fairness baselines.

These results reaffirm the main paper's observations regarding ENSUR's ability to balance fairness and performance while adapting effectively across diverse datasets.

*Table 3.* Performances and fairness comparisons with base models and fairness baselines on the **MovieLens Dataset** grouped by the **gender** in terms of risk, average set size, and Hit Rate Diff/DCG Diff, respectively. Bold indicates the best result, underline indicates the second best and † marks threshold exceeded cases.

| Method | Group | Risk ↓ | Average Set Size ↓ | Hit Rate | DCG | Hit Rate Diff ↓ | DCG Diff ↓ |
|---|---|---|---|---|---|---|---|
| DeepFM | 1 | 0.2 | | 0.8 | 0.503 | 0.017 | 0.022 |
| | 2 | 0.183 | 9 | 0.817 | 0.525 | | |
| DeepFM + ENSUR | 1 | 0.188 | | 0.812 | 0.504 | 0.002 | 0.018 |
| | 2 | 0.187 | | 0.813 | 0.522 | | |
| GMF | 1 | 0.147 | | 0.853 | 0.538 | 0.051 | 0.019 |
| | 2 | 0.198 | 9 | 0.802 | 0.519 | | |
| GMF + ENSUR | 1 | 0.155 | | 0.845 | 0.526 | 0.043 | 0.008 |
| | 2 | 0.198 | | 0.802 | 0.517 | | |
| LightGCN | 1 | 0.212 † | | 0.788 | 0.432 | 0.077 | 0.043 |
| | 2 | 0.289 † | 19 | 0.711 | 0.389 | | |
| LightGCN + ENSUR | 1 | 0.128 | | 0.873 | 0.47 | **0.001** | 0.031 |
| | 2 | 0.128 | | 0.872 | 0.44 | | |
| MLP | 1 | 0.173 | | 0.827 | 0.553 | 0.016 | 0.007 |
| | 2 | 0.158 | 7 | 0.842 | 0.56 | | |
| MLP + ENSUR | 1 | 0.151 | | 0.849 | 0.557 | 0.014 | **0.004** |
| | 2 | 0.165 | | 0.835 | 0.553 | | |
| NeuMF | 1 | 0.199 | | 0.802 | 0.542 | 0.002 | 0.015 |
| | 2 | 0.198 | 8 | 0.802 | 0.557 | | |
| NeuMF + ENSUR | 1 | 0.149 | | 0.851 | 0.556 | 0.05 | 0.005 |
| | 2 | 0.199 | | 0.801 | 0.551 | | |
| **Other Fairness Baselines** | | | | | | | |
| NFCF | 1 | 0.198 | 8 | 0.802 | 0.539 | 0.01 | 0.01 |
| | 2 | 0.205 † | | 0.795 | 0.549 | | |
| MFCF | 1 | 0.243 † | 9 | 0.757 | 0.552 | 0.002 | 0.009 |
| | 2 | 0.242 † | | 0.758 | 0.561 | | |
| NeuMF-UFR | 1 | 0.216 † | 8 | 0.784 | 0.528 | 0.034 | 0.021 |
| | 2 | 0.182 | | 0.818 | 0.549 | | |
| GMF-UFR | 1 | 0.215 † | 9 | 0.785 | 0.527 | 0.03 | 0.022 |
| | 2 | 0.185 | | 0.815 | 0.549 | | |

*Table 4.* Performances and fairness comparisons with base models and fairness baselines on the **Last.fM Dataset** grouped by the **Region** in terms of risk, average set size, and Hit Rate Diff/DCG Diff, respectively. Bold indicates the best result, underline indicates the second best and † marks threshold exceeded cases.

| Method | Group | Risk ↓ | Average Set Size ↓ | Hit Rate | DCG | Hit Rate Diff ↓ | DCG Diff ↓ |
|---|---|---|---|---|---|---|---|
| DeepFM | 1 | 0.171 | | 0.829 | 0.363 | 0.108 | 0.111 |
| | 2 | 0.279† | 29 | 0.721 | 0.252 | | |
| DeepFM + ENSUR | 1 | 0.181 | | 0.819 | 0.358 | 0.016 | 0.016 |
| | 2 | 0.197 | | 0.803 | 0.342 | | |
| GMF | 1 | 0.186 | | 0.814 | 0.268 | 0.107 | 0.071 |
| | 2 | 0.293† | 45 | 0.707 | 0.197 | | |
| GMF + ENSUR | 1 | 0.156 | | 0.844 | 0.273 | 0.019 | 0.023 |
| | 2 | 0.175 | | 0.825 | 0.25 | | |
| LightGCN | 1 | 0.217† | | 0.783 | 0.382 | 0.026 | 0.013 |
| | 2 | 0.243† | 25 | 0.757 | 0.369 | | |
| LightGCN + ENSUR | 1 | 0.164 | | 0.836 | 0.392 | 0.03 | 0.02 |
| | 2 | 0.194 | | 0.806 | 0.39 | | |
| MLP | 1 | 0.221† | | 0.779 | 0.328 | 0.019 | 0.013 |
| | 2 | 0.24† | 32 | 0.76 | 0.315 | | |
| MLP + ENSUR | 1 | 0.197 | | 0.803 | 0.331 | **0.007** | 0.008 |
| | 2 | 0.19 | | 0.81 | 0.323 | | |
| NeuMF | 1 | 0.201 † | | 0.799 | 0.323 | 0.068 | 0.021 |
| | 2 | 0.269 † | 30 | 0.731 | 0.302 | | |
| NeuMF + ENSUR | 1 | 0.187 | | 0.813 | 0.330 | 0.011 | **0.004** |
| | 2 | 0.198 | | 0.802 | 0.326 | | |
| **Other Fairness Baselines** | | | | | | | |
| NFCF | 1 | 0.248 † | 30 | 0.752 | 0.344 | 0.024 | 0.049 |
| | 2 | 0.272† | | 0.728 | 0.295 | | |
| MFCF | 1 | 0.231 † | 45 | 0.769 | 0.269 | 0.066 | 0.051 |
| | 2 | 0.297† | | 0.703 | 0.218 | | |
| NeuMF-UFR | 1 | 0.213 † | 30 | 0.787 | 0.306 | 0.045 | 0.019 |
| | 2 | 0.258 † | | 0.742 | 0.287 | | |
| GMF-UFR | 1 | 0.211 † | 45 | 0.789 | 0.245 | 0.067 | 0.048 |
| | 2 | 0.278 † | | 0.722 | 0.197 | | |

*Table 5.* Performances and fairness comparisons with base models and fairness baselines on the **Book-Crossing Dataset** grouped by the **Age** in terms of risk, average set size, and Hit Rate Diff/DCG Diff, respectively. Bold indicates the best result, underline indicates the second best and † marks threshold exceeded cases.

| Method | Group | Risk ↓ | Average Set Size ↓ | Hit Rate | DCG | Hit Rate Diff ↓ | DCG Diff ↓ |
|---|---|---|---|---|---|---|---|
| DeepFM | 1 | 0.123 | 39 | 0.873 | 0.291 | 0.302† | 0.115 |
|  | 2 | 0.429† |  | 0.571 | 0.176 |  |  |
| DeepFM + ENSUR | 1 | 0.188 |  | 0.812 | 0.251 | **0.003** | 0.02 |
|  | 2 | 0.191 |  | 0.809 | 0.231 |  |  |
| GMF | 1 | 0.187 | 35 | 0.813 | 0.277 | 0.129 | 0.115 |
|  | 2 | 0.316† |  | 0.684 | 0.162 |  |  |
| GMF + ENSUR | 1 | 0.185 |  | 0.815 | 0.268 | 0.116 | 0.05 |
|  | 2 | 0.199 |  | 0.801 | 0.232 |  |  |
| LightGCN | 1 | 0.154 | **34** | 0.846 | 0.189 | 0.217† | 0.031 |
|  | 2 | 0.371 † |  | 0.629 | 0.158 |  |  |
| LightGCN + ENSUR | 1 | 0.18 |  | 0.82 | 0.186 | 0.019 | 0.025 |
|  | 2 | 0.199 |  | 0.801 | 0.161 |  |  |
| MLP | 1 | 0.124 | 36 | 0.876 | 0.225 | 0.192 | 0.093 |
|  | 2 | 0.316† |  | 0.684 | 0.132 |  |  |
| MLP + ENSUR | 1 | 0.167 |  | 0.833 | 0.194 | 0.029 | **0.019** |
|  | 2 | 0.196 |  | 0.804 | 0.175 |  |  |
| NeuMF | 1 | 0.145 | 39 | 0.855 | 0.253 | 0.214† | 0.107 |
|  | 2 | 0.359† |  | 0.641 | 0.146 |  |  |
| NeuMF + ENSUR | 1 | 0.187 |  | 0.813 | 0.227 | 0.004 | 0.036 |
|  | 2 | 0.191 |  | 0.809 | 0.204 |  |  |
| **Other Fairness Baselines** | | | | | | | |
| NFCF | 1 | 0.216 † | 39 | 0.784 | 0.264 | 0.095 | 0.08 |
|  | 2 | 0.311† |  | 0.689 | 0.184 |  |  |
| MFCF | 1 | 0.248 † | 35 | 0.752 | 0.252 | 0.087 | 0.074 |
|  | 2 | 0.335† |  | 0.665 | 0.178 |  |  |
| NeuMF-UFR | 1 | 0.183 | 39 | 0.817 | 0.236 | 0.143 | 0.041 |
|  | 2 | 0.326 † |  | 0.674 | 0.195 |  |  |
| GMF-UFR | 1 | 0.195 † | 35 | 0.805 | 0.265 | 0.118 | 0.097 |
|  | 2 | 0.313 † |  | 0.687 | 0.168 |  |  |

### E.2. Parameters Analysis -Continued

**Effect of Risk Control Parameters $\alpha$ and $\delta$ on Prediction Set Sizes**.

Figures 6 to 8 illustrate the trends in the average prediction set size as $\alpha$ varies from 0.10 to 0.50 (in increments of 0.05), while keeping the risk confidence thresholds fixed at $\delta = 0.05, 0.10, 0.15$, using the Book-Crossing, MovieLens, and Last.fm datasets respectively. Similarly, Figures 9 to 11 present the trends in the average prediction set size as $\delta$ varies from 0.10 to 0.50 (in increments of 0.05), while keeping the confidence thresholds fixed at $\alpha = 0.15, 0.20, 0.25$, using the AmazonOffice, MovieLens, and Last.fm datasets respectively.

The observed trends in Figures 6 to 8 (variation in $\alpha$) and Figures 9 to 11 (variation in $\delta$) are consistent with the observations reported in Figure 2 (AmazonOffice dataset) and Figure 3 (Book-Crossing dataset) in the main paper. These results reinforce the consistency of our framework's behavior across different datasets and grouping methods.

**Effect of Fairness Control Parameters $\eta$ and $\hat{\delta}$ on Prediction Set Sizes** Figures 12 to 14 illustrate how the average prediction set size changes as $\eta$ varies from 0.10 to 0.50 (in increments of 0.05), while holding the fairness confidence thresholds fixed at $\hat{\delta} = 0.15, 0.20, 0.25$. These results are based on the AmazonOffice dataset, Book-Crossing and Book-Crossing datasets respectively.

In contrast, Figures 15 to 17 display the trends in prediction set size as $\hat{\delta}$ ranges from 0.10 to 0.50 (in increments of 0.05), with fixed thresholds of $\eta = 0.15, 0.20, 0.25$. These findings are based on AmazonOffice, MovieLens and Last.fm datasets respectively.

These results further validate variations in $\eta$ and $\hat{\delta}$ exhibit consistent patterns, emphasizing our framework's ability to adapt prediction set sizes effectively based on fairness constraints.

## F. Generalizablity of Grouping Methods

We validate if ENSUR is adaptable to practitioners' demands for customized user groups based on specific biases or fairness concerns relevant to their context. Specifically, we test our framework using different grouping techniques on a single dataset i.e. Last.fm by grouping users based on item interactions and grouping by both item interactions and interactions with popular items on the Last.fm dataset. The results could be found in Table 6 and Table 7. The results demonstrate that the ENSUR framework can dynamically generate prediction sets for users grouped by any condition. This is particularly useful in real-world scenarios, where different applications may have different definitions of fairness. By allowing any grouping method, the framework can support dynamic fairness criteria that can evolve with changing societal norms or organizational policies, thereby allowing practitioners to define user groups based on the specific biases or fairness concerns relevant to their context.

## G. Practical Applicability of the Framework

We now analyze the practical applicability of our framework. In real-world recommendation systems, prediction sets are often fixed to a specific size $k$ and applied uniformly across all users. This fixed size is typically determined heuristically or through trial and error, aiming to maximize the likelihood of including items that users may interact with while prioritizing and ranking items by relevance. However, this heuristic approach has several limitations:

- Fixed-size sets can lead to cognitive overload for users when the size is too large or fail to meet individual user needs when the size is too small.
- They do not account for disparities in user engagement or group fairness, potentially disadvantaging certain user groups.
- Recommending unnecessary items results in resource inefficiencies for platforms.

Our framework addresses these challenges by dynamically determining the minimum prediction set size for each user, satisfying fairness and performance guarantees with statistical confidence (e.g., 95%). This complements the existing recommender systems as we can employ an appropriate ag-

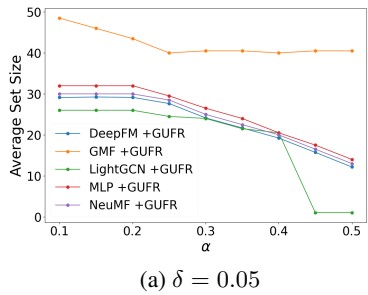 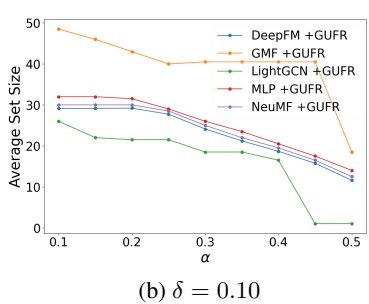 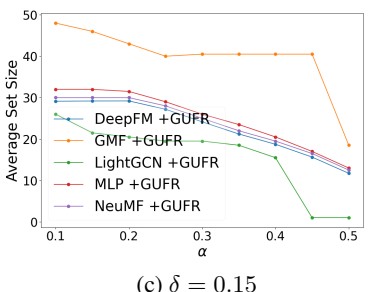

(a) $\delta = 0.05$  (b) $\delta = 0.10$  (c) $\delta = 0.15$

*Figure 6.* Analysis of base models after applying the ENSUR framework in terms of average set size with varying $\alpha = \{0.10, 0.15, 0.20, 0.25, 0.30, 0.35, 0.40, 0.45, 0.50\}$ on **Last.fm** dataset grouped by **Region** under different $\delta$.

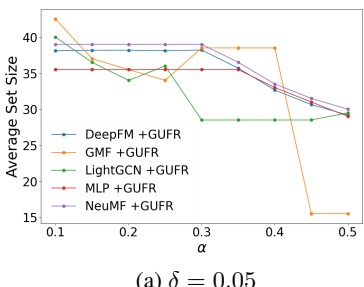 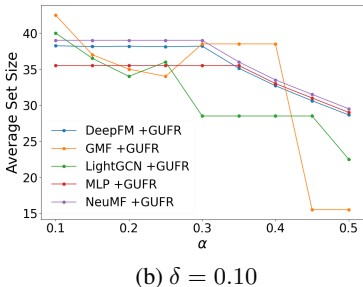 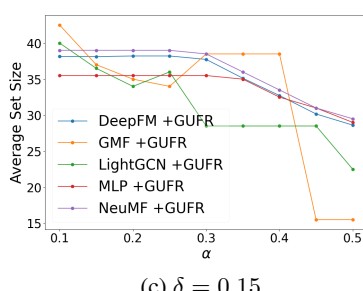

(a) $\delta = 0.05$  (b) $\delta = 0.10$  (c) $\delta = 0.15$

*Figure 7.* Analysis of base models after applying the ENSUR framework in terms of average set size with varying $\alpha = \{0.10, 0.15, 0.20, 0.25, 0.30, 0.35, 0.40, 0.45, 0.50\}$ on **Book-Crossing** dataset grouped by **Age** under different $\delta$.

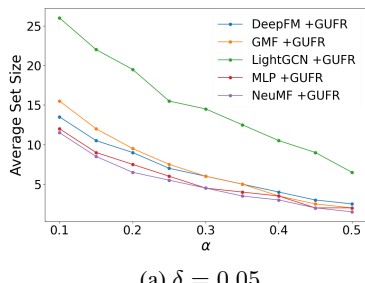 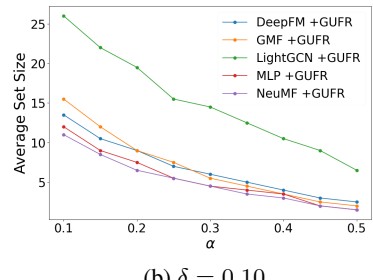 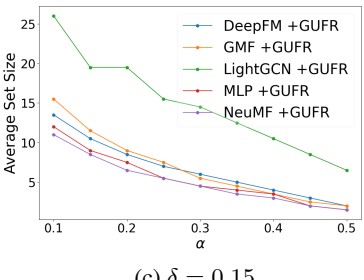

(a) $\delta = 0.05$  (b) $\delta = 0.10$  (c) $\delta = 0.15$

*Figure 8.* Analysis of base models after applying the ENSUR framework in terms of average set size with varying $\alpha = \{0.10, 0.15, 0.20, 0.25, 0.30, 0.35, 0.40, 0.45, 0.50\}$ on **MovieLens** dataset grouped by **Gender** under different $\delta$.

gregation method (for example, mean) to compute global k. This global k, obtained with the theoretical guarantees, can then be applied to recommend unseen items to users, ensuring that fairness and performance guarantees hold across the system.

For example, in e-commerce platforms such as Amazon, instead of heuristically fixing $k = 10$ for all users, our framework identifies an optimal k (e.g., $k = 7$) that balances fairness and accuracy, reducing unnecessary recommendations and enhancing user satisfaction while optimizing platform resources. Similarly, in streaming services like Netflix,

dynamically adjusting $k$ in cold-start scenarios ensures concise and personalized recommendations, preventing user overwhelm and aligning with platform resource constraints.

Additionally, the calculated $k$ can serve as a benchmark to fine-tune recommendation models, enabling iterative improvements that enhance fairness and accuracy across diverse user groups. By tailoring prediction set sizes dynamically, our framework provides a practical, scalable solution for modern recommendation systems.

.

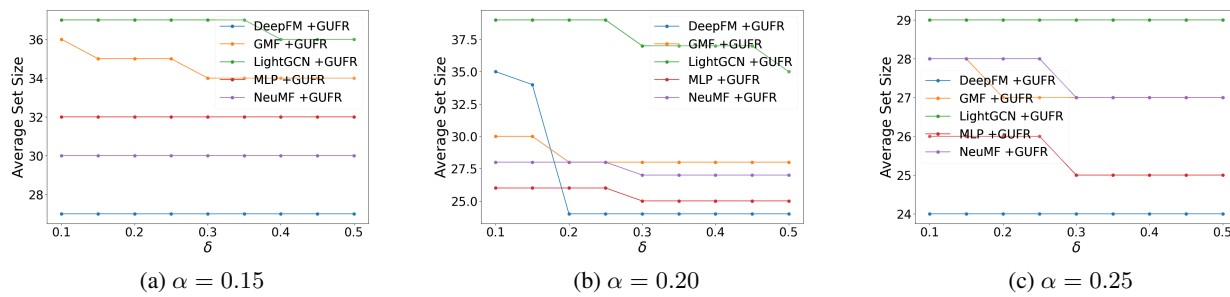

*Figure 9.* Analysis of base models after applying the ENSUR framework in terms of average set size with varying $\delta = \{0.10, 0.15, 0.20, 0.25, 0.30, 0.35, 0.40, 0.45, 0.50\}$ on **AmazonOffice** dataset grouped by **Interactions** under different $\alpha$.

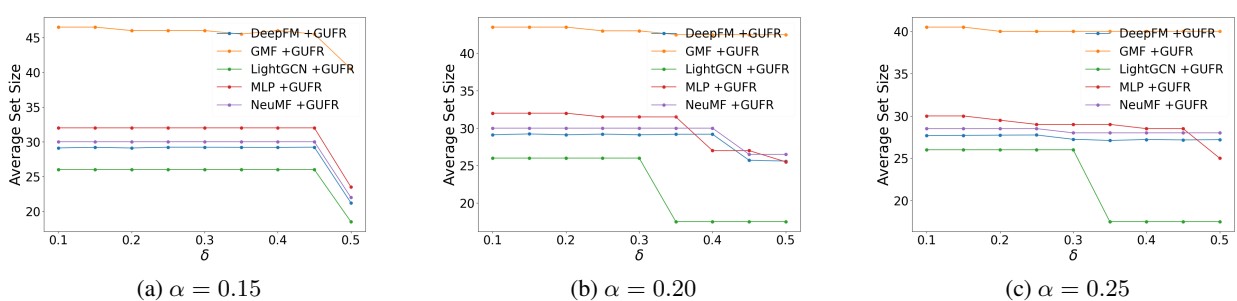

*Figure 10.* Analysis of base models after applying the ENSUR framework in terms of average set size with varying $\delta = \{0.10, 0.15, 0.20, 0.25, 0.30, 0.35, 0.40, 0.45, 0.50\}$ on **Last.fm** dataset grouped by **Region** under different $\alpha$.

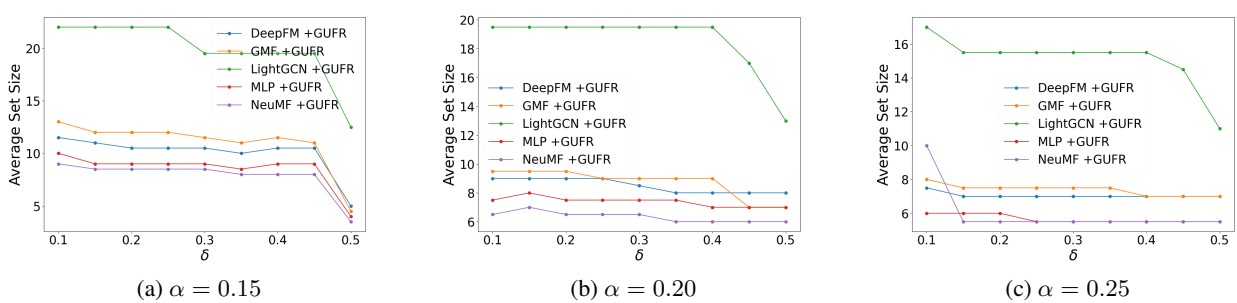

*Figure 11.* Analysis of base models after applying the ENSUR framework in terms of average set size with varying $\delta = \{0.10, 0.15, 0.20, 0.25, 0.30, 0.35, 0.40, 0.45, 0.50\}$ on **MovieLens** dataset grouped by **Gender** under different $\alpha$.

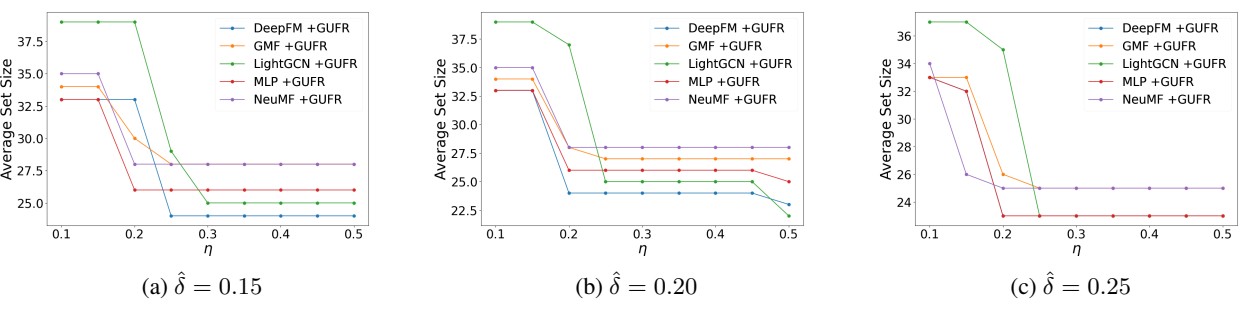

*Figure 12.* Analysis of base models after applying the ENSUR framework in terms of average set size with varying $\eta = \{0.10, 0.15, 0.20, 0.25, 0.30, 0.35, 0.40, 0.45, 0.50\}$ on **AmazonOffice** dataset grouped by **Interactions** under different $\hat{\delta}$.

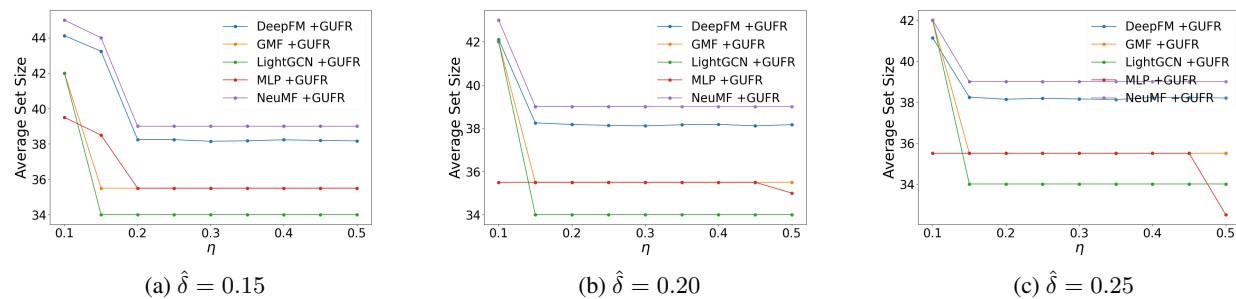

*Figure 13.* Analysis of base models after applying the ENSUR framework in terms of average set size with varying $\eta = \{0.10, 0.15, 0.20, 0.25, 0.30, 0.35, 0.40, 0.45, 0.50\}$ on **Book-Crossing** dataset grouped by **Age** under different $\hat{\delta}$.

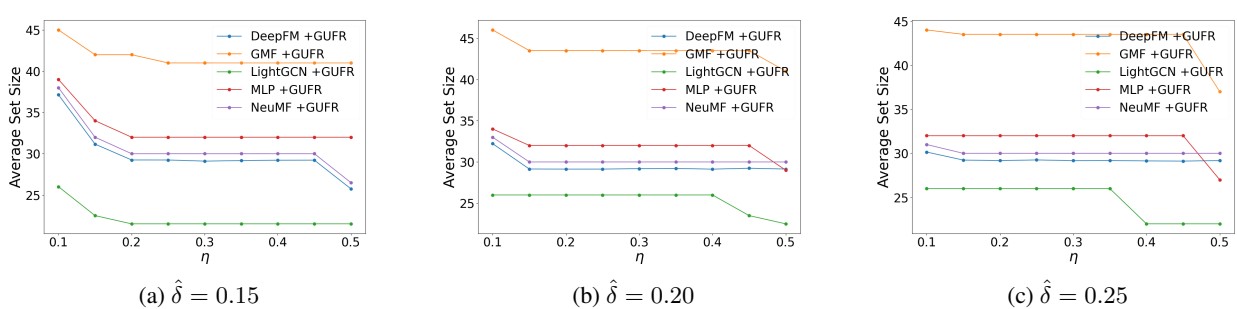

*Figure 14.* Analysis of base models after applying the ENSUR framework in terms of average set size with varying $\eta = \{0.10, 0.15, 0.20, 0.25, 0.30, 0.35, 0.40, 0.45, 0.50\}$ on **Last.fM** dataset grouped by **Region** under different $\hat{\delta}$.

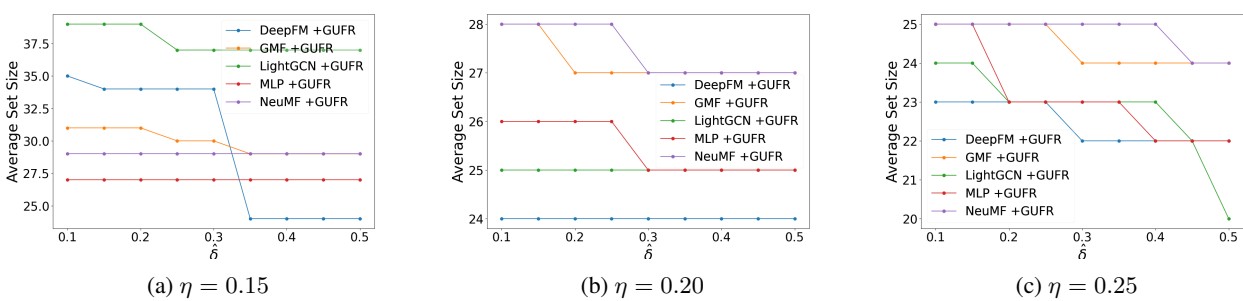

*Figure 15.* Analysis of base models after applying the ENSUR framework in terms of average set size with varying $\hat{\delta} = \{0.10, 0.15, 0.20, 0.25, 0.30, 0.35, 0.40, 0.45, 0.50\}$ on **AmazonOffice** dataset grouped by **Interactions** under different $\eta$.

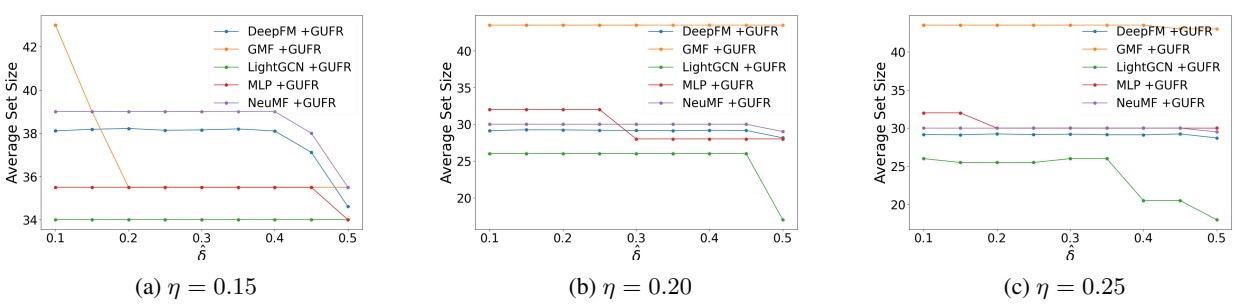

*Figure 16.* Analysis of base models after applying the ENSUR framework in terms of average set size with varying $\hat{\delta} = \{0.10, 0.15, 0.20, 0.25, 0.30, 0.35, 0.40, 0.45, 0.50\}$ on **Book-Crossing** dataset grouped by **Age** under different $\eta$.

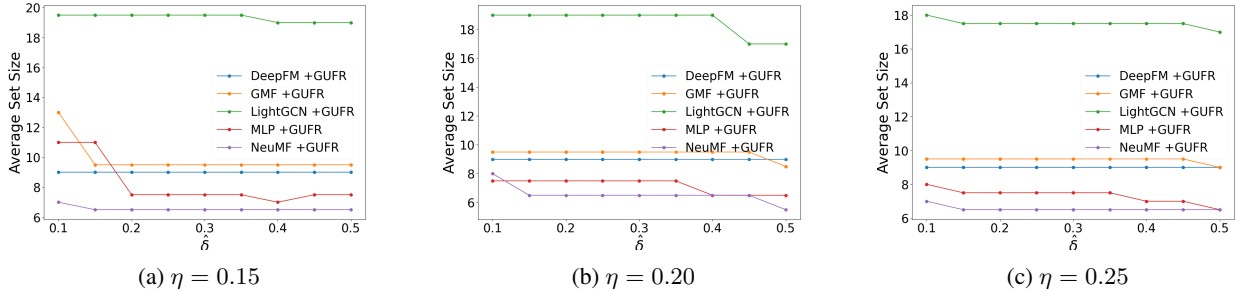

*Figure 17.* Analysis of base models after applying the ENSUR framework in terms of average set size with varying $\hat{\delta}$ = $\{0.10, 0.15, 0.20, 0.25, 0.30, 0.35, 0.40, 0.45, 0.50\}$ on **MovieLens** dataset grouped by **Gender** under different $\eta$.

*Table 6.* Performance and fairness comparisons with base models and fairness baselines on the **Last.fM Dataset** grouped by the **Item Interactions** in terms of risk, average set size, and Hit Rate Diff/DCG Diff, respectively. Bold indicates the best result, underline indicates the second best and † marks threshold exceeded cases.

| Method | Group | Risk↓ | Average Set Size ↓ | Hit Rate | DCG | Hit Rate Diff↓ | DCG Diff↓ |
|---|---|---|---|---|---|---|---|
| **Grouped by number of interactions** | | | | | | | |
| DeepFM | 1 | 0.183 | | 0.817 | 0.494 | 0.073 | 0.035 |
| | 2 | 0.255 | 16 | 0.745 | 0.458 | | |
| DeepFM + ENSUR | 1 | 0.177 | | 0.823 | 0.494 | 0.013 | **0.022** |
| | 2 | 0.19 | | 0.81 | 0.472 | | |
| GMF | 1 | 0.163 | | 0.837 | 0.567 | 0.08 | 0.066 |
| | 2 | 0.243 † | 13 | 0.757 | 0.501 | | |
| GMF + ENSUR | 1 | 0.183 | | 0.817 | 0.56 | **0.001** | 0.045 |
| | 2 | 0.183 | | 0.817 | 0.515 | | |
| LightGCN | 1 | 0.179 | | 0.821 | 0.547 | 0.18 | 0.089 |
| | 2 | 0.359† | **12** | 0.641 | 0.458 | | |
| LightGCN + ENSUR | 1 | 0.201 | | 0.799 | 0.492 | 0.003 | 0.066 |
| | 2 | 0.198 | | 0.802 | 0.426 | | |
| MLP | 1 | 0.192 | | 0.808 | 0.44 | 0.077 | 0.076 |
| | 2 | 0.269 † | 15 | 0.731 | 0.364 | | |
| MLP + ENSUR | 1 | 0.151 | | 0.849 | 0.448 | 0.041 | 0.067 |
| | 2 | 0.192 | | 0.808 | 0.38 | | |
| NeuMF | 1 | 0.151 | | 0.849 | 0.58 | 0.087 | 0.081 |
| | 2 | 0.238 † | 16 | 0.762 | 0.499 | | |
| NeuMF + ENSUR | 1 | 0.142 | | 0.858 | 0.581 | 0.027 | 0.067 |
| | 2 | 0.169 | | 0.831 | 0.513 | | |
| **Other Fairness Baselines** | | | | | | | |
| NFCF | 1 | 0.248 † | 16 | 0.822 | 0.569 | 0.039 | 0.053 |
| | 2 | 0.272† | | 0.783 | 0.516 | | |
| MFCF | 1 | 0.231 † | 13 | 0.815 | 0.529 | 0.042 | 0.021 |
| | 2 | 0.297† | | 0.773 | 0.508 | | |
| NeuMF-UFR | 1 | 0.213 † | 16 | 0.827 | 0.546 | 0.045 | 0.031 |
| | 2 | 0.258 † | | 0.782 | 0.515 | | |
| GMF-UFR | 1 | 0.211 † | 13 | 0.819 | 0.536 | 0.047 | 0.029 |
| | 2 | 0.278 † | | 0.772 | 0.517 | | |

*Table 7.* Performance, and fairness comparisons with base models and fairness baselines on the **Last.fM Dataset** grouped by the **Item Interactions & Interaction with Popular Items** in terms of risk, average set size, and Hit Rate Diff/DCG Diff, respectively. Bold indicates the best result, underline indicates the second best and † marks threshold exceeded cases.

| Method | Group | Risk↓ | Average Set Size ↓ | Hit Rate | DCG | Hit Rate Diff↓ | DCG Diff↓ |
|---|---|---|---|---|---|---|---|
| **Grouped by number of total interactions & popular items interactions** | | | | | | | |
| DeepFM | 1 | 0.078 | | 0.922 | 0.56 | 0.22† | 0.226 † |
| | 2 | 0.298† | 30 | 0.702 | 0.334 | | |
| DeepFM + ENSUR | 1 | 0.162 | | 0.838 | 0.54 | 0.162 | 0.151 |
| | 2 | 0 | | 1 | 0.389 | | |
| GMF | 1 | 0.063 | | 0.937 | 0.603 | 0.112 | 0.231 |
| | 2 | 0.176 | 30 | 0.824 | 0.372 | | |
| GMF + ENSUR | 1 | 0.163 | | 0.837 | 0.578 | 0.163 | 0.174 |
| | 2 | 0 | | 1 | 0.405 | | |
| LightGCN | 1 | 0.076 | | 0.924 | 0.627 | 0.119 | 0.24† |
| | 2 | 0.195 | 31 | 0.805 | 0.387 | | |
| LightGCN + ENSUR | 1 | 0.126 | | 0.874 | 0.585 | **0.03** | **0.106** |
| | 2 | 0.156 | | 0.844 | 0.479 | | |
| MLP | 1 | 0.057 | | 0.943 | 0.522 | 0.152 | 0.239 † |
| | 2 | 0.209† | 31 | 0.791 | 0.283 | | |
| MLP + ENSUR | 1 | 0.143 | | 0.857 | 0.5 | 0.143 | 0.179 |
| | 2 | 0 | | 1 | 0.322 | | |
| NeuMF | 1 | 0.07 | | 0.93 | 0.661 | 0.147 | 0.263 † |
| | 2 | 0.217 † | **29** | 0.783 | 0.398 | | |
| NeuMF + ENSUR | 1 | 0.154 | | 0.846 | 0.64 | 0.154 | 0.198 |
| | 2 | 0 | | 1 | 0.442 | | |
| **Other Fairness Baselines** | | | | | | | |
| NFCF | 1 | 0.115 | 29 | 0.885 | 0.629 | 0.08 | 0.208† |
| | 2 | 0.195 | | 0.805 | 0.421 | | |
| MFCF | 1 | 0.137 | 30 | 0.863 | 0.549 | 0.109 | 0.151 |
| | 2 | 0.243† | | 0.757 | 0.44 | | |
| NeuMF-UFR | 1 | 0.111 | 29 | 0.889 | 0.588 | 0.077 | 0.166 |
| | 2 | 0.188 | | 0.812 | 0.428 | | |
| GMF-UFR | 1 | 0.121 | 30 | 0.879 | 0.566 | 0.057 | 0.198 |
| | 2 | 0.178 | | 0.822 | 0.368 | | |

