# OpenReview forum: "ENSUR: Equitable and Statistically Unbiased Recommendation"
_ICML.cc/2025/Conference — ICML 2025 poster_

### Official Review · Reviewer_at4n · 2025-03-12

**Overall Recommendation:** 4

**Summary:**

This paper introduces ENSUR (Equitable and Statistically Unbiased Recommendation), a novel framework aimed at ensuring confidence and fairness in recommender systems. The authors propose a dynamic method for generating prediction sets that guarantee:
1.	A user-predefined confidence level (e.g., 90%) for including the true item,
2.	Fairness across different user groups,
3.	Minimal average prediction set sizes.
To achieve these goals, the authors develop the Guaranteed User Fairness Algorithm (GUFA), which optimizes fairness and risk constraints efficiently. They establish theoretical guarantees for fairness control, risk control, and minimal prediction set size. Extensive experiments validate the ENSUR framework across multiple datasets and base models, demonstrating improved fairness and reliability without sacrificing performance.

**Claims And Evidence:**

The paper’s claims are well-supported by rigorous theoretical analysis and empirical validation. The authors provide detailed proofs for fairness and risk control guarantees (Theorem 5.1) and minimal prediction set size (Theorem 5.2). Additionally, they derive upper bounds for risk and fairness metrics to accelerate optimization (Theorem 4.1 and Theorem 4.2). The experimental results across four datasets confirm the framework’s effectiveness, showing improved fairness and risk control compared to baseline methods.

**Essential References Not Discussed:**

The paper covers essential prior work, particularly in fairness-aware recommendation and statistical risk control. No significant omissions were identified.

**Experimental Designs Or Analyses:**

The experimental setup is comprehensive, covering four diverse datasets (e.g., MovieLens, AmazonOffice) and five base recommendation models. The comparisons with fairness baselines (NFCF, MFCF, GMF-UFR, NeuMF-UFR) are appropriate, and the results consistently support the authors’ claims. The parameter analysis further enhances the credibility of the findings. The efficiency comparison demonstrates ENSUR’s computational advantage over existing fairness methods.

**Methods And Evaluation Criteria:**

The proposed methodology is well-founded and aligns with established fairness-aware recommendation frameworks. The authors adapt the Risk-Controlling Prediction Sets (RCPS) approach while incorporating fairness constraints, making their contributions novel and practically relevant. The evaluation criteria—risk control, fairness control, and minimal prediction set size—are well-justified, and the results demonstrate meaningful improvements over baseline models.

**Other Comments Or Suggestions:**

1.	It would be useful to discuss potential limitations when fairness groups are highly imbalanced.

2.	Consider elaborating on real-world applicability beyond academic datasets.

3.	Future work could explore alternative fairness constraints and their implications.

**Other Strengths And Weaknesses:**

Strengths:

•	Theoretical rigor: Provides strong mathematical guarantees for fairness, risk control, and efficiency.

•	Practical relevance: Demonstrates applicability across multiple datasets and base models.

•	Computational efficiency: ENSUR outperforms fairness baselines in training time.

•	Clear writing and well-structured methodology.

Weaknesses:

•	Assumptions: Some fairness constraints may not hold universally across all domains.

•	Evaluation scope: While diverse datasets are used, real-world deployment studies would further strengthen the impact.

**Questions For Authors:**

1.	How does ENSUR perform when fairness groups have significantly different sizes? Does it remain stable under severe imbalance?

2.	Could the fairness constraints be extended to multi-group settings beyond binary group definitions?

3.	Are there practical limitations to applying ENSUR in online recommendation scenarios?

**Relation To Broader Scientific Literature:**

This paper extends prior work on fairness in recommendation (e.g., Yao & Huang, 2017; Abdollahpouri et al., 2019) and uncertainty quantification via risk-controlling prediction sets (e.g., Bates et al., 2021). ENSUR builds upon these foundations by integrating fairness and confidence guarantees into a unified optimization framework. The approach is novel in its combination of theoretical guarantees and practical implementation, making it a valuable contribution to fairness-aware recommendation research.

**Theoretical Claims:**

The theoretical claims in this paper are sound and rigorously proved. Theorems 4.1 and 4.2 provide upper bounds for risk and fairness metrics, facilitating efficient optimization. Theorems 5.1 and 5.2 ensure fairness and risk constraints while maintaining minimal prediction set size. The derivations are mathematically solid, and the assumptions are well-motivated and clearly stated.

---

> ### Author Rebuttal · Authors · 2025-03-31
>
> We sincerely thank the reviewer for their encouraging feedback and for appreciating the relevance, rigor, and practical efficiency of our framework. Below, we address the questions:
>
> a) ENSUR's performance under significant group size imbalance:
> While the empirical results presented in the paper cover moderately imbalanced scenarios, ENSUR remains robust under more severe imbalances. This is primarily due to its design: each user group's risk and fairness thresholds are independently optimized. As a result, ENSUR ensures stable statistical guarantees—even when groups differ significantly in size—by tailoring the learned calibration parameter ($\lambda$) to each group’s distribution. However, underrepresented groups may require more conservative λ values, leading to slightly larger prediction sets. We will clarify this behavior in the Discussion in the revised version.
>
> b) Extending fairness constraints to multi-group settings:
> Thank you for this insightful question. The optimization in GUFA can effectively handle multiple fairness constraints simultaneously by introducing a unique fairness threshold (η) for each group. Each group's constraint will correspond to its threshold ($\eta$), and the optimization problem remains structurally the same. For example, if a dataset has three user subgroups (e.g., by age, gender, and region), GUFA jointly calibrates all three using a separate fairness constraint. This makes the extension to multi-group fairness both practical and computationally efficient.
>
> c) Practical limitations in applying ENSUR to online recommendation scenarios:
> We appreciate this critical point. ENSUR, as a statistical calibration framework, remains practically applicable to online recommendation scenarios, but given dynamic changes in user behavior online, ENSUR would require periodic recalibration (e.g., daily, weekly, or monthly, depending on system dynamics and data drift). Given ENSUR's efficiency, such recalibration is feasible in large-scale systems. We will expand this point further in the Discussion section and explore this direction in our future work.
>
> Once again, we are incredibly grateful to the reviewers for their valuable comments, suggestions, and positive recognition of our work.

---

### Official Review · Reviewer_BirQ · 2025-03-12

**Overall Recommendation:** 3

**Summary:**

The paper introduces ENSUR (Equitable and Statistically Unbiased Recommendation), a framework designed to enhance fairness and confidence in recommender systems. The core idea is to generate dynamic prediction sets that (1) ensure a high-confidence inclusion of the true item, (2) guarantee fairness across diverse user groups, and (3) minimize set sizes for efficiency. To achieve this, the authors propose the Guaranteed User Fairness Algorithm (GUFA), which optimizes prediction sets while maintaining statistical risk and fairness bounds. The paper provides a rigorous theoretical foundation, derives upper bounds for risk and fairness control, and supports claims with extensive empirical evaluations.

**Claims And Evidence:**

Yes.

**Essential References Not Discussed:**

No.

**Experimental Designs Or Analyses:**

The experimental evaluation is comprehensive, covering multiple datasets (AmazonOffice, MovieLens, Last.fm, Book-Crossing) and a variety of base models (DeepFM, GMF, MLP, NeuMF, LightGCN).

**Methods And Evaluation Criteria:**

Yes.

**Other Comments Or Suggestions:**

No.

**Other Strengths And Weaknesses:**

Novel contribution: The combination of risk control and fairness guarantees in recommendation is innovative. Rigorous theoretical foundation: Well-structured proofs and upper-bound derivations.Strong empirical validation: Comprehensive experiments demonstrate real-world applicability. Computational efficiency: ENSUR is significantly faster than existing fairness-aware baselines.

Fairness group assumptions: The approach assumes predefined user groups, which may not always be straightforward in practice.

**Questions For Authors:**

1. How does ENSUR adapt when fairness groups are not predefined but inferred dynamically from user interactions?
2. Could ENSUR be extended to handle multi-sided fairness constraints (e.g., fairness for both users and content providers)?
3. Are there any observed limitations when applying ENSUR to highly imbalanced user groups?

**Relation To Broader Scientific Literature:**

The authors draw on concepts from risk-controlling prediction sets (Bates et al., 2021) and fairness-aware collaborative filtering (e.g., Yao & Huang, 2017; Abdollahpouri et al., 2019).

**Theoretical Claims:**

Theorem 4.1 and Theorem 4.2 establish upper bounds for risk and fairness, which are correct.

---

> ### Author Rebuttal · Authors · 2025-03-31
>
> We thank the reviewer for their thoughtful review and for recognizing our framework's novelty, theoretical rigor, and practical efficiency. We appreciate their insightful comments and questions and address them below:
>
> a) Adaptation of ENSUR when fairness groups are inferred dynamically:
> We thank the reviewer for the important point. While ENSUR assumes predefined user groups, we can naturally extend it to dynamically inferred groups using clustering techniques based on user interaction patterns (e.g., click history, content preferences, time of activity).  We can rerun the ENSUR's statistical calibration step periodically or upon significant distributional changes in user behavior, reflecting these evolving group definitions. Since GUFA performs per-group calibration and is computationally efficient, such updates are feasible at scale. This approach ensures that fairness and confidence guarantees are preserved even as the definitions of user groups evolve over time. In the final version of the paper, we will add a Discussion on  this adaptation strategy.
>
> b) Extension to multi-sided fairness constraints
> We are thankful for suggesting this interesting direction. Our framework can be generalized to multi-sided fairness by adding multiple fairness constraints simultaneously— i.e., fairness for users as well as content providers. Each side (user or provider) will have associated fairness thresholds ($\eta$ values), and the GUFA optimization strategy will jointly ensure fairness guarantees are maintained across all these dimensions while ensuring that the combined objective remains tractable. This extension maintains the modularity of the current approach and is practically beneficial in many multi-stakeholder platforms such as marketplaces.
>
> c) Limitations when applying ENSUR to highly imbalanced user groups:
> We sincerely appreciate this critical observation. In highly imbalanced settings, ENSUR will continue to maintain valid fairness guarantees as the calibration of risk and fairness constraints is done independently for every group. This will ensure statistical robustness. However, in such settings, due to underrepresentation, the minority group may require a more conservative learned parameter $\lambda$ to ensure statistical guarantees. This may eventually result in larger prediction set sizes for underrepresented groups. We recognize that extreme imbalance is a real-world challenge and will discuss ENSUR behavior under such cases and its implications in the Discussion section of the revised version.
>
> We are again deeply grateful to the reviewer for their positive acknowledgment of our work and for their valuable questions.

---

### Official Review · Reviewer_xCED · 2025-03-12

**Overall Recommendation:** 4

**Summary:**

This paper proposes a novel and reliable framework called Equitable and Statistically Unbiased Recommendation (ENSUR)) to dynamically generate prediction sets for users across various groups. This paper further designs an efficient algorithm named Guaranteed User Fairness Algorithm (GUFA) to optimize the proposed method and derive upper bounds of risk and fairness metrics to speed up optimization process. Rigorous theoretical analysis and extensive experiments are also provided.

## update after rebuttal

Authors have addressed my concerns.

**Claims And Evidence:**

Yes

**Essential References Not Discussed:**

In my view, related works are currently cited/discussed in this paper.

**Experimental Designs Or Analyses:**

Yes. I check the setting of experiments in Section 6.1.

**Methods And Evaluation Criteria:**

Yes

**Other Comments Or Suggestions:**

Some minor issues: Eq.(12) in line 226.

**Other Strengths And Weaknesses:**

Strengths:

a. This paper is well-written and easy to follow.

b. The proposed method and theory in this paper is solid.

c. Unlike existing frameworks, this paper offers rigorous theoretical guarantees.

d. It is highly efficient compared to current baselines.

e. The framework is lightweight and is model and dataset-agnostic.

f. Comprehensive experiments are conducted across multiple datasets, models, and user groupings. Extensive hyperparameters experiments are done to show how they impact coverage, performance and fairness.

Weakness:

a. I appreciate the proposed method and theory, but in my view, the technique in method or proof process of theorems is not surprising.

b. The authors could explain further on how the choice of hyperparameters is made in the main paper.

c. The framework illustration could be elaborated further to make a reader understand the work quickly.

**Questions For Authors:**

a. I need a complete overview of the technical innovations of this paper. In order to prove the theory in the article, what technique is used in this article, what problems does this technique solve, and how innovative is this technique?

b. How should the practitioners determine appropriate fairness thresholds (eta)?

c. How does guaranteeing a minimum average prediction set size improve recommendation quality? Some real-world example will be helpful

**Relation To Broader Scientific Literature:**

The paper builds on Risk-Controlling Prediction Sets (Bates et al., 2021) and fairness-aware recommendation literature.

**Theoretical Claims:**

I checked the proofs of this paper and do not find the errors.

---

> ### Author Rebuttal · Authors · 2025-03-31
>
> We are grateful to the reviewer for their encouraging and supportive feedback. We are glad they found our paper well-written, rigorous, and efficient. Below, we aim to address their insightful queries and suggestions:
>
> a) Technical Innovation:
> Our technical innovation lies in ENSUR being a unified statistical framework for generating recommendation sets that satisfy both confidence and group fairness constraints. We achieve this by formulating the recommendation problem into the Risk-Controlling Prediction Sets (RCPS) and Fairness-Constrained Prediction Sets (FCPS) paradigm, thereby integrating them into a single calibration process. A key innovation is the development of the GUFA (Guaranteed User Fairness Algorithm), a powerful yet efficient algorithm that enables post hoc calibration without retraining the base recommender model. GUFA leverages theoretically derived upper bounds on risk and fairness violations (presented in Theorems 4.1 and 4.2), using derived Binomial and Bernstein concentration inequalities to make the optimization tractable. These bounds are further used to construct guarantees for risk and fairness control in Theorems 5.1 and 5.2 while ensuring minimal prediction set size.
> While prior methods usually rely on heuristic or empirical approaches, our approach thereby results in a model and dataset-agnostic pipeline that is theoretically sound, computationally efficient, and practically deployable, with no need for any architecture-specific modifications or retraining.
>
> b) Selection of Fairness Threshold ($\eta$) by practitioners:
> We thank the reviewer for the insightful question. To effectively choose a fairness threshold (η), the practitioners can: 1) assess historical performance disparity between user groups. 2) conduct preliminary analyses on validation datasets to balance fairness requirements against recommendation performance, and 3) finally align the thresholds with institutional fairness standards. We will add a detailed discussion about the approach to paper revision. For example, in the AmazonOffice Dataset, we set $\eta = 0.2\$ by selecting the smallest value satisfying fairness constraints across groups based on interaction count, using our defined fairness metrics verified on validation sets.
>
> c) Real-world Value of Minimizing Prediction Set Size:
> Minimizing average prediction set size significantly improves recommendation quality as: 1) It reduces cognitive overload on users, thereby reducing the problem of ad blindness 2) On a streaming platform, a minimum yet accurate prediction set could help in skipping prefetching and caching trailers for irrelevant content thereby saving bandwidth and compute. 3) a smaller yet better prediction set size in ads will lead to higher click-through rates (CTR) and better returns.
>
> We also thank the reviewer for highlighting the minor issue in Eq. (12) and suggesting improving our framework illustration. We will address them in the final revision.
>
> Once again, we thank the reviewer for their thoughtful questions and encouragement, and we look forward to strengthening the final version of the paper based on the valuable feedback.

---

### Official Review · Reviewer_zLvX · 2025-03-14

**Overall Recommendation:** 4

**Summary:**

The paper introduces a comprehensive framework named ENSUR (Ensuring Statistical Fairness and Confidence in Recommendation Systems), which is designed to statistically ensure both fairness and confidence in the outcomes generated by recommendation systems. The authors propose that by utilizing two key components, RCPS (Recommendation Confidence Probability Sets) and FCPS (Fairness Constrained Probability Sets), the framework can generate high-confidence dynamic recommendation sets tailored to individual users. These sets not only meet the stringent requirements of confidence levels but also rigorously satisfy group fairness criteria, ensuring that the recommendations are equitable across different groups.

**Claims And Evidence:**

Yes

**Essential References Not Discussed:**

I think the essential references are discussed in the paper.

**Experimental Designs Or Analyses:**

The experimental designs are thorough and sound.

**Methods And Evaluation Criteria:**

Yes

**Other Comments Or Suggestions:**

No.

**Other Strengths And Weaknesses:**

The proposed framework, ENSUR, is designed to be both model-agnostic and dataset-agnostic, meaning it can seamlessly integrate with any base recommender model and adapt to diverse datasets across various domains without requiring specific modifications or constraints. One of the key strengths of ENSUR is its ability to provide rigorous theoretical guarantees for fairness, a critical aspect that has been notably absent in prior works, thereby addressing a significant gap in the field of fair recommendation systems. To validate the effectiveness of the proposed framework, the authors conduct extensive and comprehensive experiments that span multiple datasets and compare against numerous fairness baselines. Additionally, the study includes thorough and complete hyperparameter sensitivity testing, which meticulously examines the impact of different parameter settings on the framework's performance, further solidifying the reliability and generalizability of the results. The experimental outcomes demonstrate that ENSUR achieves significant improvements over existing baselines, not only in terms of recommendation performance but also in computational efficiency, highlighting its practical utility and superiority in real-world applications. Overall, the framework's versatility, theoretical rigor, and empirical validation make it a substantial advancement in the pursuit of fair and efficient recommendation systems.


While the paper presents a robust and well-structured framework, it could benefit from providing more detailed explanations regarding the selection and tuning of hyperparameters within the main body of the text, particularly addressing how these choices might vary across different dataset settings and domains. Such insights would offer readers a clearer understanding of the practical considerations involved in implementing the framework and how to adapt it to specific use cases. Additionally, while the supplementary material is comprehensive and contains valuable technical details, its density and complexity might make it less accessible to readers who are not deeply familiar with the mathematical foundations of the work. To improve readability and accessibility, the authors could consider summarizing the key proof ideas and theoretical insights in the main paper, using intuitive explanations and high-level overviews to convey the core concepts without overwhelming readers who may not have a strong mathematical background. This approach would make the paper more inclusive and engaging for a broader audience, while still preserving the rigor and depth of the technical content for experts in the field.

**Questions For Authors:**

No.

**Relation To Broader Scientific Literature:**

The paper extends the prior works in fairness in recommendation systems.

**Theoretical Claims:**

I think there is no problem.

---

> ### Author Rebuttal · Authors · 2025-03-31
>
> We are thankful to the reviewer for their positive and encouraging feedback. We sincerely appreciate their recognition of our framework's rigor, versatility, and substantial empirical validation.
>
> a) Hyperparameter Selection and Tuning:
> We acknowledge a detailed explanation presenting hyperparameter selection and tuning will significantly benefit the paper's clarity.  The choice of our hyperparameter parameters (e.g., risk threshold $\alpha$, fairness threshold $\eta$, confidence parameters $\delta$ and $\hat{\delta}$) is primarily guided by extensive empirical validation and standard practices from relevant literature such as Bates et al. (2021).  For example, on the AmazonOffice Dataset, the risk threshold ($\alpha$) is set as 0.2  to control over-coverage while ensuring robustness, and η is chosen as 0.2 by selecting the smallest value satisfying fairness constraints across groups, based on interaction count using defined fairness metrics, verified on validation sets. Similarly, the confidence parameters δ and $\hat{\delta}$ were chosen from the set $\{0.1, 0.15, 0.2, 0.25, 0.3, 0.35, 0.4, 0.45, 0.5}\$ based on the value that consistently achieved the desired confidence level without unnecessarily large recommendation sizes. We followed this same procedure across the other datasets. We will include a detailed discussion of these guidelines in the final version.
>
> b) We thank the reviewer for their valuable suggestion to improve readability. In the final version, we will include intuitive explanations of the theoretical contributions in Section 5 alongside the formal results. Similar to the existing remarks for Theorems 5.1 and 5.2, we will explain how Theorems 4.1 and 4.2 provide a high-probability surrogate for population-level constraints and how these bounds are later used in Theorems 5.1 and 5.2, which further guide the design of the GUFA optimization procedure. We will illustrate this flow clearly for accessibility.
>
> Based on the valuable feedback, we will (i) add a dedicated paragraph in Section 6 explaining hyperparameter selection across datasets, supported by our sensitivity experiments from Section 6.2.2, and (ii) give intuitive summaries of the main theoretical results for broader understanding.
>
> Again, we are deeply thankful to the reviewer for the positive outlook of our paper and their insightful suggestions for further improving its accessibility and clarity.

---

### Decision · Program_Chairs · 2025-05-01

**Decision:**

Accept (poster)

**Comment:**

The paper proposes a new framework ENSUR to statistically ensure both fairness and confidence in recommendation systems.  All the reviewers think that this paper did a good job in deriving the rigorous theoretical guarantees for fairness and agree to accept this paper.